# Enhancing Lensless Imaging via Explicit Learning of Model Mismatch

## Abstract

Emerging lensless imaging techniques hold promise for miniaturized cameras, but their effectiveness is constrained by challenges like model mismatch from the point spread function (PSF), which undermines reconstruction methods dependent on accurate PSF modeling. To address this issue, we propose a joint Maximum a Posteriori (MAP) approach to simultaneously estimate model mismatch error ($M^2E$) and reconstruct high-resolution images from lensless imaging measurements. Specifically, we propose an explicit latent space representation for $M^2E$ to improve robustness against PSF inaccuracies. Additionally, we develop a multi-stage reconstruction network by unfolding the joint MAP estimator with a learned Laplacian Scale Mixture (LSM) prior and $M^2E$ representation ($M^2ER$) through end-to-end optimization. Extensive experiments show that our method surpasses current state-of-the-art methods.

## 1 Introduction

Lensless imaging is integral to inverse imaging research, offering compact, budget-friendly camera solutions (Pan et al. (2022); Lee et al. (2023)). Differing significantly from traditional optical methods, lensless imaging encodes information as diffraction patterns and uses computational methods for lensless image reconstruction (Zuo et al. (2024)), as shown in Fig. 1 (a). However, the presence of misalignment, lateral shift, object-to-sensor distance (OSD) variations, and environmental / system noise in point spread function (PSF) of the lensless imaging system brings the model mismatch error ($M^2E$) for lensless image reconstruction method (Zeng & Lam (2021); Yang et al. (2022); Li et al. (2023); Qian et al. (2024)), as shown in Fig. 1 (b).

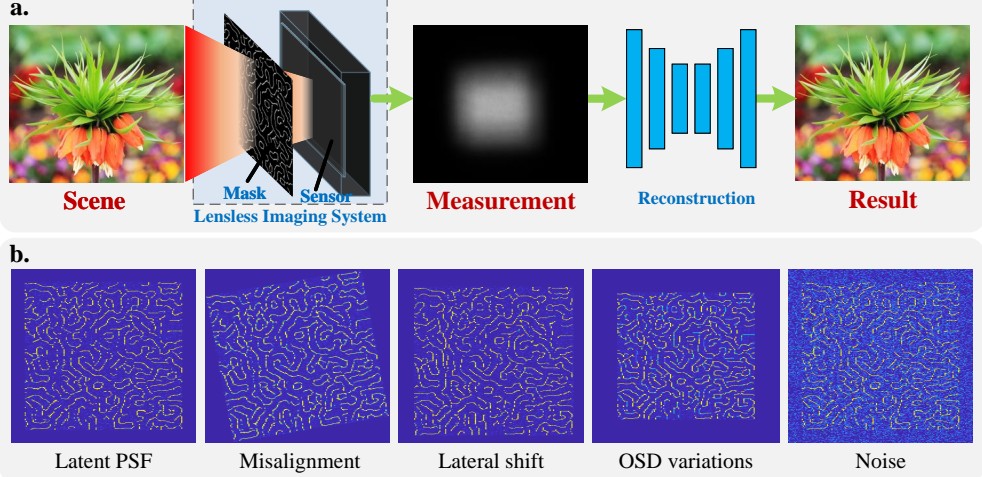

Figure 1: Brief pipeline of lensless imging and reconstrcution (a) and causes of $M^2E$ (b).

Most of existing reconstruction techniques in lensless imaging utilize data-driven methods such as the multi-stage networks (Wu et al. (2021)), generative adversarial learning frameworks (Salman

et al. (2022); Lee et al. (2023)), Transformers (Pan et al. (2022)), and diffusion model-based methods (Wan et al. (2023)). They fail to integrate knowledge of the forward imaging model, overlook $M^2E$, and are limited by their reliance on specific system setups, making them less adaptable. Minor changes in these setups (*i.e.*, minor $M^2E$) requires retraining, hindering generalizability across different imaging scenarios.

Some studies leverage physical priors to image degradation caused by the $M^2E$. Such as, a PSF-learned deep unfolding method (Yang et al. (2022)) is proposed to joint of image reconstrcution and denoise. Furthermore, Wiener deconvolution within a multiscale feature space (Li et al. (2023)) is employed to enhance input correction, effectively minimizing information loss and mitigating $M^2E$. In another study (Banerjee & Singh (2024)), multiple PSFs are utilized to develop a sparse convolutional PSF-aware auxiliary branch, enabling CycleGAN to mitigate $M^2E$ and enhance reconstruction accuracy. These methods rely on learning-based corrections for inputs or PSFs without providing explicit representations from physical models, limiting their ability to effectively eliminate $M^2E$.

Recent studies (Zeng & Lam (2021); Qian et al. (2024)) have characterized $M^2E$ as an additive bias within the latent model, conceptualizing it as a specific noise through unfolding and merging operators. To mitigate its effects, these studies have incorporated specialized denoising mechanisms aimed at reducing $M^2E$. Although these methods involve the $M^2E$, they do not explicitly integrate it into the computational framework, primarily serving as correction schemes.

To fundamentally improve lensless image reconstruction affected by $M^2E$, we explicitly quantify the $M^2E$ and integrate it into a co-optimization framework. Specifically, we propose a co-learning network by converting the joint Maximum a Posteriori (MAP) estimator with a learned Laplacian Scale Mixture (LSM) prior and estimated $M^2E$ into a multi-stage deep unfolding network. In a nutshell, our contributions are listed as follows:

- The lensless image reconstruction task is first formulated as a joint MAP method for co-estimating the $M^2E$ and reconstructing the underlying scene. We propose a $M^2E$ learning-aware reconstruction network called as $M^2LNet$ by incorporating the MAP estimator with a learned LSM prior and estimated $M^2E$ into a multi-stage reconstruction framework in an end-to-end learning manner.

- An explicit learning model called as $M^2E$ representation ($M^2ER$) is proposed to improve the robustness of $M^2E$ estimation. Both the feature (mean) and uncertainty (variance) in the latent space of the $M^2E$ are learned, aided in the learning of $M^2E$.

- Extensive experiments on datasets captured by two prototypes, PHlatCam and our Fin-Cam, demonstrate that our method can significantly improve lensless image reconstruction performance and has the potential to be applied to other lensless cameras.

## 2 RELATED WORKS

### 2.1 LENSLESS IMAGING

Lensless imaging systems (M. Salman et al. (2017); Nick et al. (2018); Pan et al. (2022); Wu et al. (2020); Adams et al. (2022)), which replace bulky lenses with thin optical masks, are emerging as a compact alternative to traditional cameras. These systems use amplitude (M. Salman et al. (2017); Pan et al. (2022)) or phase masks (Nick et al. (2018); Wu et al. (2020)) to project light diffusely onto the sensor, requiring advanced algorithms to decode the captured scene. Recent prototypes, such as FlatCam (M. Salman et al. (2017)), DiffuserCam (Nick et al. (2018)), PHlatCam (Boominathan et al. (2020)), and FZA-based cameras (Wu et al. (2020; 2021)), have demonstrated significant improvements in imaging quality through enhanced reconstruction algorithms.

The growing advantages of lensless imaging have driven its adoption in ultrafast optical, hyperspectral, and microscopic imaging. Studies like (Zhao & Li (2022)) and (Touil et al. (2022)) achieved single-shot ultrafast optical imaging by combining an acoustic-optic programmable dispersive filter with spectrally filtered time all-optical mapping. For hyperspectral imaging (Monakhova et al. (2020)), a compact computational camera uses a spectral filter array on the sensor and a nearby diffuser. Additionally, a scatter-plate microscope (Alok et al. (2017)) leverages random medium diffusion for diffraction-limited microstructure imaging. In vivo tissue imaging (Adams et al. (2022)) with a phase mask produced a high-contrast PSF covering a broad spatial frequency range. Recent works (Pan et al. (2021); Yin et al. (2022)) further explored object inference using lensless cameras,

emphasizing their versatility across many applications. As technology progresses, lensless cameras play crucial roles in compact, lightweight, and computationally advanced imaging solutions.

## 2.2 IMAGE RECONSTRUCTION FOR LENSLESS IMAGING

The advances in deep learning have notably impacted computational imaging, particularly lensless imaging (Sinha et al. (2017); Salman et al. (2022); Wu et al. (2021)). Models such as UNet (Sinha et al. (2017)) and its variants (Horisaki et al. (2020)) have been adapted for lensless image reconstruction, while GANs (Rego et al. (2021); Ni et al. (2024); Banerjee & Singh (2024)) have been employed to improve visual fidelity by estimating the single PSF or muilt-PSF. Recently, Transformer-based method (Pan et al. (2022)) is proposed for leveraging long-range dependencies to enhance reconstruction. These models analyze extensive datasets to find correlations between lensless measurements and corresponding scenes. However, the presence of inappropriate data can significantly impair reconstruction quality.

Recent studies (Yang et al. (2019); Zhao et al. (2022b); Dong et al. (2023)) have explored integrating model-based methods with deep learning networks. For instance, in (Monakhova et al. (2019)), authores combined unrolled ADMM with UNet denoisers for lensless image reconstruction. Although these methods improve reconstruction performance, their reliance on accurate imaging model and minor $M^2E$ limits practical use. To address this, a PSF-learned deep unfolding strategy (Yang et al. (2022)) to mitigate $M^2E$, as well as, Wiener deconvolution operator within a multiscale feature space (Li et al. (2023)) is employed to reduce $M^2E$. Latest studies (Zeng & Lam (2021); Qian et al. (2024)) shows that characterizing $M^2E$ as an additive bias within the latent model helps lensless image reconstruction. However, they do not adequately address how to fundamentally suppress $M^2E$. Unlike the aforementioned methods, our method distinguishes itself by explicitly addressing $M^2E$ in lensless imaging. Our method models and corrects M2E using a novel latent space representation, and integrate LSM prior in a joint MAP framework, enabling more accurate reconstruction.

Unlike the aforementioned methods, our method mitigates the impact of $M^2E$ by explicitly modeling it and incorporating this consideration during reconstruction.

## 3 METHODOLOGY

To enhance clarity of this paper, this section initially presents the problem formulation (Sec. 3.1), followed by an in-depth discussion on $M^2E$ modeling and its network architecture (Sec. 3.2). Subsequently, we explore the integration of $M^2E$ with a multi-stage lensless imaging reconstruction network (Sec. 3.3). Finally, we give the a comprehensive description of the overall framework called $M^2$LNet (Sec. 3.4), as depicted in Fig. 2.

## 3.1 PROBLEM FORMULATION

According to (Ni et al. (2024)), the lensless imaging measurement $\mathbf{y}$ can be modeled as:

$$\mathbf{y} = \Phi \circledast \mathbf{x} + \mathbf{n} = \mathcal{O}\mathbf{x} + \mathbf{n} \tag{1}$$

where $\circledast$ represents convolution operation, $\mathbf{x}$ denotes the underlying scene, and $\mathbf{n}$ is the noise. Note that we default the system matrix $\mathcal{O}$ as the agent of PSF $\Phi$ to unify the description. Thus, the model mismatch and PSF mismatch are equivalent.

The forward imaging model described in Eq. (29) allows computable modeling of lensless image reconstruction, but it requires an accurate PSF. In practice, the on-axis PSF obtained from experimental measurements or simulations based on mask patterns and imaging geometry may contain significant deviation against the ground truths, thus leading to the model mismatch that would bring substantial artifacts in the reconstructed images.

For this, we introduce the model mismatch denoted as $\Delta_{\mathcal{O}}$ to represent the mismatch between the biased PSF and the actual PSF. As a result, we have the following lensless imaging forward model:

$$\mathbf{y} = \left( \hat{\mathcal{O}} + \Delta_{\hat{\mathcal{O}}} \right) \mathbf{x} + \mathbf{n}, \tag{2}$$

where $\hat{\mathcal{O}}$ and $\mathcal{O} = \hat{\mathcal{O}} + \Delta_{\hat{\mathcal{O}}}$ are biased and true one, respectively.

Figure 2: The architecture of our $\text{M}^2\text{LNet}$. It consists of a trainable fidelity reconstruction (TFR) module, a $\text{M}^2\text{E}$ respresentation ($\text{M}^2\text{ER}$) module, and a multi-stage reconstruction network (MSRN). The MSRN comprises several cascaded stages with DPMB.

Consequently, according to Taylor expansion, the lensless image reconstruction by inversion operation (Zeng & Lam (2021)) can be written as

$$
\begin{aligned}
\hat{\mathbf{x}} = \hat{\mathcal{O}}^{-1}\mathbf{y} &= \left(\boldsymbol{I} - \mathcal{O}^{-1}\Delta_{\hat{\mathcal{O}}}\right)^{-1}\left(\mathbf{x} + \mathcal{O}^{-1}\mathbf{n}\right) \\
&= \left(\boldsymbol{I} + \mathcal{O}^{-1}\Delta_{\hat{\mathcal{O}}}\right)\left(\mathbf{x} + \mathcal{O}^{-1}\mathbf{n}\right) + o\left(\left\|\Delta_{\hat{\mathcal{O}}}\right\|_F^2\right) \\
&= \left(\boldsymbol{I} + \mathcal{O}^{-1}\Delta_{\hat{\mathcal{O}}}\right)\mathbf{x} + \left(\boldsymbol{I} + \mathcal{O}^{-1}\Delta_{\hat{\mathcal{O}}}\right)\mathcal{O}^{-1}\mathbf{n} + o\left(\left\|\Delta_{\hat{\mathcal{O}}}\right\|_F^2\right) \\
&= \mathcal{A}\mathbf{x} + \xi,
\end{aligned}
\tag{3}
$$

where $\mathcal{A}$ is the $\text{M}^2\text{E}$ formulated as

$$
\mathcal{A} = \boldsymbol{I} + \mathcal{O}^{-1}\Delta_{\hat{\mathcal{O}}} = \boldsymbol{I} + (\hat{\mathcal{O}} + \Delta_{\hat{\mathcal{O}}})^{-1}\Delta_{\hat{\mathcal{O}}}.
\tag{4}
$$

And the $\xi = \mathcal{A}(\hat{\mathcal{O}} + \Delta_{\hat{\mathcal{O}}})^{-1}\mathbf{n} + o\left(\left\|\Delta_{\hat{\mathcal{O}}}\right\|_F^2\right)$ represents the mixed interference under the influence of measurement noise and $\text{M}^2\text{E}$. The $\boldsymbol{I}$ is the identity matrix. $\|\cdot\|_F^2$ denotes Frobenius norm.

Lensless image reconstruction involves estimating $\mathcal{A}$ and recovering $\mathbf{x}$ from $\hat{\mathbf{x}}$ and $\mathcal{A}$, posing a highly ill-posed inverse problem. We formulate it as maximum posteriori (MAP) estimation:

$$
p(\mathcal{A}, \mathbf{x}|\hat{\mathbf{x}}) = p(\mathcal{A}|\hat{\mathbf{x}})\, p(\mathbf{x}|\mathcal{A}, \hat{\mathbf{x}}) = p(\mathcal{A}|\hat{\mathbf{x}})\frac{p(\hat{\mathbf{x}}|\mathcal{A}, \mathbf{x})p(\mathbf{x})}{p(\hat{\mathbf{x}}|\mathcal{A})},
\tag{5}
$$

where $p(\hat{\mathbf{x}}|\mathcal{A}) = \int p(\hat{\mathbf{x}}|\mathcal{A}, \mathbf{x})p(\mathbf{x})d\mathbf{x}$ is a normalization constant ensuring the proper normalization of the conditional probability. Ignoring this term and taking logarithms on both sides of equation,

$$
\log p(\mathcal{A}, \mathbf{x}|\hat{\mathbf{x}}) \propto \log p(\mathcal{A}|\hat{\mathbf{x}}) + \log p(\hat{\mathbf{x}}|\mathcal{A}, \mathbf{x}) + \log p(\mathbf{x}),
\tag{6}
$$

then solving the MAP problem can be expressed as

$$
(\mathcal{A}^*, \mathbf{x}^*) = \underset{\mathcal{A}, \mathbf{x}}{\arg\max}\, \log p(\mathcal{A}|\hat{\mathbf{x}}) + \log p(\hat{\mathbf{x}}|\mathcal{A}, \mathbf{x}) + \log p(\mathbf{x}).
\tag{7}
$$

where $\mathcal{A}^*$ and $\mathbf{x}^*$ are the expected value of $\mathcal{A}$ and $\mathbf{x}$, respectively. According to (Zhao et al. (2022a)), the Eq. (7) can be converted into two subproblems:

$$
\mathcal{A}^* = \underset{\mathcal{A}}{\arg\max}\, \log p(\mathcal{A}|\hat{\mathbf{x}}),
\tag{8a}
$$

$$
\mathbf{x}^* = \underset{\mathbf{x}}{\arg\max}\, \log p(\hat{\mathbf{x}}|\mathcal{A}, \mathbf{x}) + \log p(\mathbf{x}).
\tag{8b}
$$

where Eq. (8a) denotes the estimation of $\mathcal{A}$ and Eq. (8b) represents reconstructing underlying scene from coarse image induced by model mismatch and estimated $\mathcal{A}$.

## 3.2 EXPLICIT LEARNING OF $\text{M}^2\text{E}$

Due to the effective modeling the randomness and uncertainty introduced by misalignment, OSD variations, and system noise, we employ the Gaussian distribution (Zhao et al. (2022a)) to model $\text{M}^2\text{E}$. Thus the $\Delta_{\hat{\mathcal{O}}}$ is model by the following distribution:

$$
\Delta_{\hat{\mathcal{O}}} \sim \mathcal{N}(\mu(\hat{\mathbf{x}}), \sigma^2(\hat{\mathbf{x}})).
\tag{9}
$$

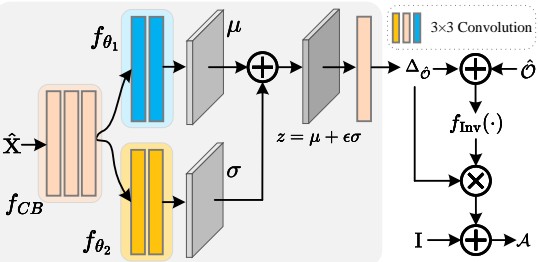

Figure 3: The architecture of $\mathrm{M^2ER}$.

By combining with Eq.( 4), thus the likelihood term $p(\mathcal{A}|\hat{\mathbf{x}})$ can be formulated as

$$p(\mathcal{A}|\hat{\mathbf{x}}) \sim \boldsymbol{I} + (\hat{\mathcal{O}} + \mathcal{N}(\mu(\hat{\mathbf{x}}), \sigma^2(\hat{\mathbf{x}})))^{-1} \mathcal{N}(\mu(\hat{\mathbf{x}}), \sigma^2(\hat{\mathbf{x}})), \tag{10}$$

where $\mu(\hat{\mathbf{x}})$ and $\sigma^2(\hat{\mathbf{x}})$ represent the mappings from $\hat{\mathbf{x}}$ to the posterior distribution parameters ($\mu$ and $\sigma$) of $\mathcal{A}$. Direct computation of these mappings is challenging, so we parameterize them as deep networks: $\mu = f_{\theta_1}(\hat{\mathbf{x}})$ and $\sigma = f_{\theta_2}(\hat{\mathbf{x}})$, where $\theta_1$ and $\theta_2$ are the parameters for the $\mu$ and $\sigma$ branches, respectively. Specifically, the coarse reconstructed image $\hat{\mathbf{x}}$ is fed into a CNN block consisting of three $3 \times 3$ convolution layers (*i.e.*, $f_{\mathrm{CB}}$) to extract feature maps of the $\mathrm{M^2E}$. These features are passed through two $3 \times 3$ convolution layers to simultaneously learn the $\mu$ and $\sigma$ of the prediction. Additionally, $\mu$ can be viewed as the identity mapping of the blur kernel, while $\sigma$ reflects the uncertainty in the predicted $\mu$. An equivalent sampling representation $z$ is then generated via the re-parameterization method:

$$z = \mu + \epsilon\sigma, \quad \epsilon \sim \mathcal{N}(\mathbf{0}, \boldsymbol{I}), \tag{11}$$

where $\epsilon$ represents random noise sampled from a normal distribution. Since $\mu$ is perturbed by $\sigma$ during training, $z$ becomes a non-deterministic embedding. However, without constraints on the embeddings, the model tends to predict a small $\sigma$ for all samples to suppress unstable components. To address this, we incorporate a Kullback-Leibler (KL) divergence regularization term (Chang et al. (2020)) to enforce a normal distribution constraint:

$$\mathcal{L}_{\mathrm{kl}} = \mathrm{KL}\left[\mathcal{N}\left(\mu, \sigma^2\right) \| \mathcal{N}(\mathbf{0}, \boldsymbol{I})\right] = -\frac{1}{2}\left(1 + \log\sigma^2 - \mu^2 - \sigma^2\right). \tag{12}$$

According to above description, we integrate $\mathcal{A}$ into a neural network framework for characterization

$$\mathcal{A} \leftarrow f_{\mathrm{M^2ER}}(\hat{\mathcal{O}}) = \boldsymbol{I} + f_\delta\left(\hat{\mathbf{x}}\right) \otimes f_{\mathrm{Inv}}\left(\hat{\mathcal{O}} + f_\delta\left(\hat{\mathbf{x}}\right)\right), \tag{13}$$

where $\mathcal{A}$ is learning-tuned, and $f_{\mathrm{M^2ER}}(\cdot)$ is the neural operator characterizing $\mathrm{M^2E}$ call $\mathrm{M^2E}$ representation ($\mathrm{M^2ER}$) module, as shown in Fig. 3. The $f_\delta(\cdot)$ explicitly maps $\Delta_{\hat{\mathcal{O}}}$ as $\Delta_{\hat{\mathcal{O}}} = \mathrm{Conv}_{3\times3}(z)$, $Conv_{3\times3}$ is the $3 \times 3$ convolution layer. $\otimes$ is a matrix-multiplication operator. $f_{\mathrm{Inv}}(\cdot)$ is an inverse operation can be described as $f_{\mathrm{Inv}}(a) = U_a\Sigma_a^{-1}V_a^\top$, and $U_a\Sigma_aV_a^\top = \mathrm{SVD}(a)$.

### 3.3 Multi-Stage Lensless Image Reconstruction

**LSM Model for Lensless Image Reconstruction.** To solve Eq. (8b), we note that $p(\hat{\mathbf{x}}|\mathcal{A}, \mathbf{x})$ is the likelihood term and $p(\mathbf{x})$ is the prior distribution of $\mathbf{x}$. The likelihood term can be generally modeled by a Gaussian distribution

$$p(\hat{\mathbf{x}}|\mathcal{A}, \mathbf{x}) = \frac{1}{\sqrt{2\pi}\sigma_n} \exp\left(-\frac{\|\hat{\mathbf{x}} - \mathcal{A}\mathbf{x}\|_2^2}{2\sigma_n^2}\right). \tag{14}$$

To effectively model the sparsity and edge characteristics inherent in natural images, we propose to characterize each pixel $x_i$ with a nonzero-mean Laplacian distribution of mean $v_i$ and variance $2\omega_i^2$:

$$p\left(x_i|\omega_i\right) = \frac{1}{2\omega_i} \exp\left(-\frac{|x_i - v_i|}{\omega_i}\right). \tag{15}$$

With the assumption that $x_i$ and $\omega_i$ are independent, we can model $\mathbf{x}$ with the following LSM model

$$p(\mathbf{x}) = \prod_i p(x_i), \quad p(x_i) = \int_0^\infty p(x_i|\omega_i) p(\omega_i) d\omega_i, \tag{16}$$

where the scale prior $p(\omega_i)$ can be modeled by a general energy function $p(\omega_i) \propto \exp(-J(\omega_i))$. Then Eq. (8b) is equivalent to a bivariate estimation problem

$$(\mathbf{x}^*, \omega^*) = \underset{\mathbf{x}, \omega}{\arg\max} \log p(\hat{\mathbf{x}}|\mathcal{A}, \mathbf{x}) + \log p(\mathbf{x}|\omega) + \log p(\omega). \tag{17}$$

By substituting the Gaussian likelihood term of Eq. (14), the prior terms of Eq. (15) into the MAP estimator in Eq. (17), we can obtain the following objective function

$$(\mathbf{x}^*, \omega^*) = \underset{\mathbf{x}, \omega}{\arg\min} \frac{1}{2}\|\hat{\mathbf{x}} - \mathcal{A}\mathbf{x}\|_2^2 + \sum_{i=1}^N \frac{\sigma_n^2}{\omega_i}|x_i - \upsilon_i| + \mathbf{\Omega}(\omega), \tag{18}$$

where $\mathbf{\Omega}(\omega) = \sigma_n^2 \sum_{i=1}^N \log \omega_i + \sigma_n^2 J(\omega)$, $J(\omega)$ is regularization term on $\omega$. Then the lensless image reconstruction problem can be solved by alternating optimizing $\mathbf{x}$ and $\omega$. For the $\mathbf{x}$-subproblem, with fixed $\omega$, we can solve $\mathbf{x}$ by

$$\mathbf{x}^* = \underset{\mathbf{x}}{\arg\min} \frac{1}{2}\|\hat{\mathbf{x}} - \mathcal{A}\mathbf{x}\|_2^2 + \sum_{i=1}^N \varsigma_i|x_i - \upsilon_i|, \tag{19}$$

where $\varsigma_i = \frac{\sigma_n^2}{\omega_i}$. Inspired by recent advances in image denoising (Zhang et al. (2022)), the mean $\upsilon_i$ can be predicted by a deep denoising module, *i.e.* $\upsilon_i = f_d(x_i)$, where $f_d(\cdot)$ denotes a denoiser. Then the Eq. (19) can be solved by the iterative shrinkage thresholding algorithm as

$$\mathbf{x}^{(k+1)} = \mathcal{S}_{\boldsymbol{\tau}^{(k)}, \boldsymbol{\upsilon}^{(k)}}\left(\mathbf{x}^{(k)} + \frac{1}{c}\mathcal{A}^\top\left(\hat{\mathbf{x}} - \mathcal{A}\mathbf{x}^{(k)}\right)\right) \tag{20}$$

where $c$ is chosen to ensure convergence. $\mathcal{S}_{\boldsymbol{\tau}^{(k)}, \boldsymbol{\upsilon}^{(k)}}(\cdot)$ denotes a generalized shrinkage operator with threshold $\boldsymbol{\tau}^{(k)} = \frac{\varsigma_i^{(k)}}{c}$ and $\boldsymbol{\upsilon}^{(k)}$, which is defined by

$$\mathcal{S}_{\boldsymbol{\tau}, \boldsymbol{\upsilon}}(\boldsymbol{t}) = \begin{cases} \boldsymbol{t} + \boldsymbol{\upsilon}, & \boldsymbol{t} < \boldsymbol{\upsilon} - \boldsymbol{\tau} \\ \boldsymbol{\upsilon}, & \boldsymbol{\upsilon} - \boldsymbol{\tau} \leq \boldsymbol{t} \leq \boldsymbol{\upsilon} + \boldsymbol{\tau} \\ \boldsymbol{t} - \boldsymbol{\upsilon}, & \boldsymbol{t} > \boldsymbol{\upsilon} + \boldsymbol{\tau} \end{cases} \tag{21}$$

Similarly, the $\omega$-subproblem is equivalent to solve the $\varsigma$-subproblem. With a fixed $\mathbf{x}$, we have

$$\varsigma^* = \underset{\varsigma}{\arg\min} \sum_{i=1}^N \varsigma_i|x_i - \upsilon_i| + \mathbf{\Omega}(\varsigma). \tag{22}$$

Functional optimization method (Yang et al. (2022)) can be used to solve $\varsigma$, which depends on a hand-crafted prior $p(\omega)$ in $\mathbf{\Omega}(\varsigma)$. Instead of using a fixed prior, we propose to estimate $\varsigma^{(k)}$ from $\hat{\mathbf{x}}^{(k)}$ by a designed DPMB, as detailed in Appendix. A.3.

**Multi-stage network for Lensless Image Reconstruction.** Despite the theoretical rigor, alternatively solving $\mathbf{x}$ and $\varsigma$ requires many iterations to converge and needs a hand-crafted prior $p(\omega)$. Meanwhile, all parameters and the denoiser can not be jointly optimized. To address these issues, we replace all variables in Eq. (20) with a common expression containing $\mathbf{x}$, so that $\mathbf{x}$ and $\varsigma$ can be jointly optimized in a unified framework as

$$\mathbf{x}^{(k+1)} = \mathcal{S}_{\frac{\mathcal{G}_\varsigma(\mathbf{x}^{(k)})}{c}, \mathcal{G}_{\boldsymbol{\upsilon}}(\mathbf{x}^{(k)})}\left(\mathbf{x}^{(k)} + \frac{1}{c}\mathcal{A}^\top\left(\hat{\mathbf{x}} - \mathcal{A}\mathbf{x}^{(k)}\right)\right). \tag{23}$$

### 3.4 COMPREHENSIVE NETWORK ARCHITECTURE

The comprehensive network architecture is shown in Fig. 2, which consists of a trainable fidelity reconstruction (TFR) module, a $\mathrm{M}^2\mathrm{ER}$ module, and a multi-stage reconstruction network (MSRN). The TFR module is design for obtaining coarse image by performing a Hadamard product in the

Fourier domain, along with a least-squares operation, denoted as $\hat{\mathbf{x}} = \left( \overline{\mathcal{F}}^{-1} \operatorname{diag} \left[ \overline{\mathcal{F}} \left( \mathbf{\Phi} \right) \right] \overline{\mathcal{F}} \right)^{-1} \mathbf{y}$, which is the networked form of Eq. (3). The $\mathrm{M}^2\mathrm{ER}$ module, as defined in Eq.(13), is responsible for mining $\mathcal{A}$. Each stage in the MSRN directly aligns with steps in the optimization process, executing $K$ iterations of Eqs. (22) and (23) with the input of $\mathcal{A}$ and $\hat{\mathbf{x}}$. Eq.(22) functions as a denoiser, implemented with the proposed DPMB, which follows with an encoder-decoder architecture to estimate the weight $\varsigma^{(k)}$ and mean $\upsilon^{(k)}$, as shown in Fig. 12 of Apppendix A.3. The K-th output corresponding to Eq. (23) regards the final reconstruction result.

## 3.5 LOSS FUNCTION

We impose supervision on predictions of each stage by MSE loss (Yang et al. (2022)), perceptual loss (Yang et al. (2022)), and KL loss (Chang et al. (2020)). Our total loss is written as

$$\mathcal{L}_{\text{all}} = \mathcal{L}_{\text{mse}} + \lambda_1 \mathcal{L}_{\text{P}} + \lambda_2 \mathcal{L}_{\text{kl}}, \tag{24}$$

where $\lambda_1$ and $\lambda_2$ are set to $0.01$ and $0.1$, respectively.

## 4 EXPERIMENTS AND RESULTS

### 4.1 DATASETS

The datasets are captured by two prototypes, PHlatCam (Boominathan et al. (2020)), and our FinCam, forming PHlatCam Display Captured Dataset (DCD-PHlatCam), and FinCam Display Captured Dataset (DCD-FinCam), respectively.

**DCD-PHlatCam.** The DCD-PHlatCam dataset is the public dataset gerenated from a subset of the ILSVRC 2012 dataset (Russakovsky et al. (2015)) for fair evaluation. The images are first resized to $384 \times 384$ and displayed on a monitor for imaging. PHlatCam, equipped with a 12.2 MP Sony IMX226 sensor, then captures lensless measurements at a resolution of $1280 \times 1480$ pixels. The dataset is split into two parts: a training set with 9900 images and a testing set with 100 images.

**DCD-FinCam.** The DCD-FinCam dataset is based on a subset of ImageNet. Paired lensless measurements are captured using our custom-built FinCam (Fig. 10 in Apppendix). The images are resized to $320 \times 320 \times 3$ as ground truths and projected onto an LCD. FinCam captures the lensless measurements, which are converted to Bayer data at $1024 \times 1536 \times 4$. The dataset includes 9900 pairs for training and 100 for testing.

### 4.2 SETUPS

**Evaluation Metrics.** We use the peak-signal-to-noise ratio (PSNR), the structural similarity index (SSIM), and the learned perceptual image patch similarity (LPIPS) metrics to assess the performance of various methods. Additionally, the number of parameters (#Param), floating point operations per second (FLOPs), and frames per second (FPS) are used for evaluating computational complexity.

**Implementation Bodies.** For training, we use the Adam optimizer with "cos" learning rate scheduling policy: $lr = 0.5 \times init\_r \times (1 + cos(\pi * epoch/max\_epoch))$, the initial learning rate ($init\_r$) is set to $5 \times 10^{-4}$, and the maximum number of epochs ($max\_epoch$) is 100. The whole network is trained with a batch size of 8. We use the Pytorch framework on a Linux 20.04 server with single NVIDIA GTX3090 GPU for all experiments.

### 4.3 COMPARISONS WITH STATE-OF-THE-ARTS ON PHLATCAM

We assess the performance on DCD-PHlatCam by comparing the reconstruction results with measured PSF. We present a comparative analysis between our $\mathrm{M}^2\mathrm{LNet}$ and several cutting-edge data-driven methods, including UDN (Banerjee et al. (2023)), UNet (Horisaki et al. (2020)), MMCN Zeng & Lam (2021), ULAMP-Net Yang et al. (2022), and MDGAN (Ni et al. (2024)). The results are shown in Fig. 4 and Tab. 1. In Fig. 4, the visual reconstruction performance of $\mathrm{M}^2\mathrm{LNet}$ is superior compared with state-of-the-art methods. Additionally, Tab. 1 presents the quantitative comparison results. Our $\mathrm{M}^2\mathrm{LNet}$ outperforms all other state-of-the-art models, consistent with the visual reconstruction performance.

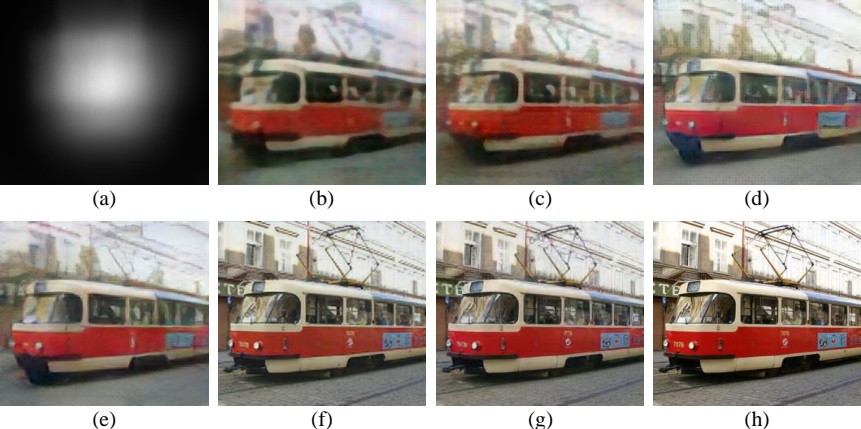

(a)  (b)  (c)  (d)

(e)  (f)  (g)  (h)

Figure 4: Visual inspection of the reconstruction performance for DCD-PHlatCam by (b) UDN (Banerjee et al. (2023)), (c) UNet (Horisaki et al. (2020)), (d) MMCN Zeng & Lam (2021), (e) ULAMP-Net Yang et al. (2022), (f) MDGAN (Ni et al. (2024)), and (g) our M$^2$LNet. (a) is the lensless imaging measurements corresponding to (h) ground truths.

Table 1: Comparison of reconstructed performance on DCD-PHlatCam. The best 1-st,2-nd,3-rd results are shown in red, green, and blue.

| Method | PSNR (dB) ↑ | SSIM ↑ | LPIPS ↓ |
|---|---|---|---|
| UDN (Banerjee et al. (2023)) | 14.11 | 0.2927 | 0.6237 |
| UNet (Horisaki et al. (2020)) | 18.83 | 0.4503 | 0.3617 |
| MMCN Zeng & Lam (2021) | 20.44 | 0.5401 | 0.3472 |
| ULAMP-Net Yang et al. (2022) | 22.28 | 0.6097 | 0.2835 |
| MDGAN (Ni et al. (2024)) | 22.59 | 0.6142 | 0.2782 |
| M$^2$LNet (ours) | 23.63 | 0.7527 | 0.2649 |

### 4.4 COMPARISONS WITH STATE-OF-THE-ARTS ON FINCAM

The visual comparisons in Fig. 5 shows that our M$^2$LNet outperforms state-of-the-art methods (*i.e.*, UDN, UNet, MMCN, ULAMP-Net, and MDGAN) with superior image quality. M$^2$LNet provides more accurate colors and textures, closely matching the ground truths, while other methods show color biases. It also produces sharper boundaries and clearer textures than ULAMP-Net and MDGAN. Table 2 shows the quantitative results on the DCD-FinCam dataset. Our M$^2$LNet leads with a PSNR of 24.19 dB, SSIM of 0.7566, and LPIPS of 0.2533, outperforming MDGAN, which scores 23.69 dB, 0.6203, and 0.2621. This marks a 2.1% improvement in PSNR, 22.0% in SSIM, and 3.4% in LPIPS. While MDGAN benefits from adversarial learning, it demands more computational resources. Due to the consideration of M$^2$E, our M$^2$LNet achieves superior reconstruction. Furthermore, we evaluate the robustness of M$^2$LNet by using histograms for PSNR, SSIM, LPIPS, and their standard deviations (Fig. 6). M$^2$LNet consistently outperforms across all metrics and shows lower standard deviations, indicating stable and robust reconstruction. Additionally, we present further experimental comparison results in Apppendix A.4 and A.5.

Table 2: Comparison of reconstructed performance on DCD-FinCam. The best 1-st,2-nd,3-rd results are shown in red, green, and blue.

| Method | PSNR (dB) ↑ | SSIM ↑ | LPIPS ↓ | FLOPs (G) ↓ | #Param (M) ↓ | FPS ↑ |
|---|---|---|---|---|---|---|
| UDN (Banerjee et al. (2023)) | 15.43 | 0.3289 | 0.5824 | 17.81 | 2.20 | 5.50 |
| UNet (Horisaki et al. (2020)) | 19.35 | 0.4763 | 0.3548 | 119.90 | 59.40 | 36.99 |
| MMCN (Zeng & Lam (2021)) | 20.44 | 0.5487 | 0.3307 | 365.50 | 206.14 | 10.68 |
| ULAMP-Net (Yang et al. (2022)) | 23.61 | 0.6182 | 0.2674 | 29.24 | 3.01 | 38.35 |
| MDGAN (Ni et al. (2024)) | 23.69 | 0.6203 | 0.2621 | 492.30 | 507.52 | 12.99 |
| M$^2$LNet (ours) | 24.19 | 0.7566 | 0.2533 | 277.41 | 343.20 | 18.92 |

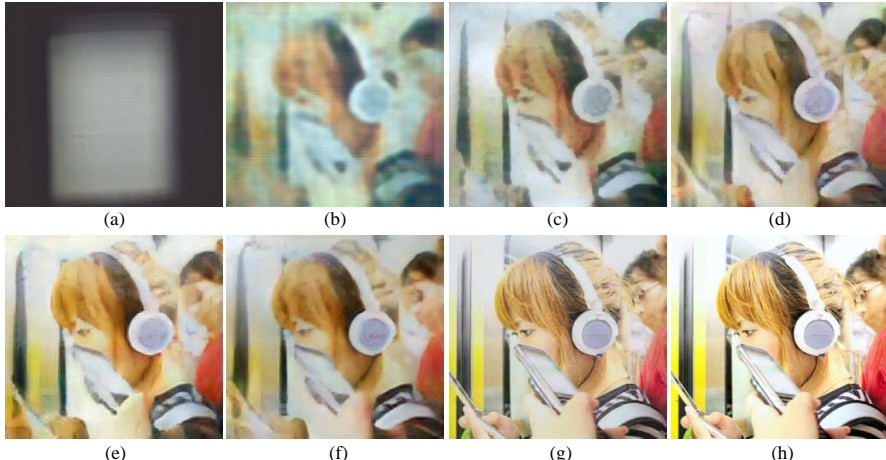

Figure 5: Visual inspection of the reconstruction performance for DCD-FinCam by (b) UDN (Banerjee et al. (2023)), (c) UNet (Horisaki et al. (2020)), (d) MMCN (Zeng & Lam (2021)), (e) ULAMP-Net (Yang et al. (2022)), (f) MDGAN (Ni et al. (2024)), and (g) our $M^2$LNet. (a) is the lensless imaging measurements corresponding to ground truths (h).

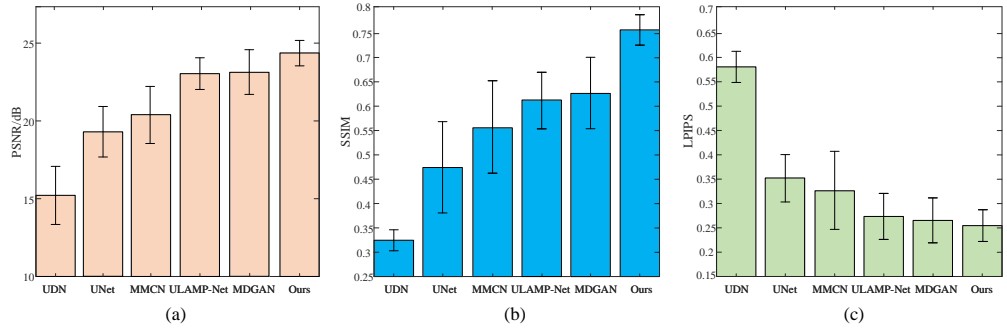

Figure 6: Illustration of the robustness of our method and other state-of-the-art methods in terms of PSNR, SSIM, LPIPS and their standard deviations on DCD-FinCam.

### 4.5 COMPLEXITY ANALYSIS

Tab. 2 presents the complexity comparison results between the above compared methods and our $M^2$LNet in terms of the #Param, FLOPs, and FPS metrics. Due to the the multi-stage reconstruction strategy used for improving accuracy at the expense of complexity, our $M^2$LNet has a relatively high computational complexity and ranks in the middle of all compared methods. Furthermore, the FPS still reaches 18.92 for FinCam, slightly below the real-time operational requirements. In the future, we will work on modeling simplified designs to improve operational efficiency.

### 4.6 ABLATION STUDIES

To simplify this work, our ablation experiments are all studied on the DCD-PHlatCam dataset. Some experimental results can be found in Apppendix A.6.

**The Accuracy of $M^2$E Prediction.** To thoroughly investigate this, we manually inject mix biases generating by the combination of translating and rotating to the PSF for simulating the biased PSF, and then through the Eq. (2), we obtain the corresponding simulation datasets, on which we train our $M^2$LNet. Here, we present a comparison between the learned $M^2$E ($\mathcal{A}$) and the true $M^2$E $(I + (\hat{\mathcal{O}} + \Delta_{\hat{\mathcal{O}}})^{-1}\Delta_{\hat{\mathcal{O}}})$, as shown in Fig. 7. The visualization results show that the $\mathcal{A}$ learned by our method closely align with the true $M^2$E.

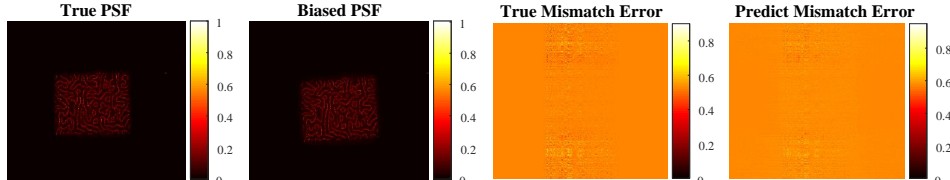

Figure 7: Visualization prediction results of $M^2E$. The $M^2E$ predicted by our method highly matches the true $M^2E$.

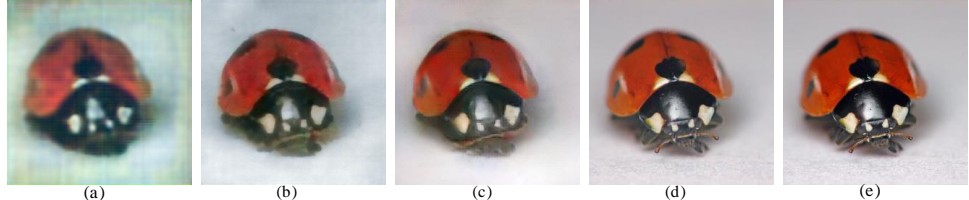

Figure 8: Visual results of ablation study on components. (a)–(d) bind to $\#\mathrm{Conf}_1$–$\#\mathrm{Conf}_4$. (e) is ground truth.

**Ablation Studies on Components.** The experiments evaluate the effect of removing individual components from $M^2$LNet on reconstruction performance. TFR+MSRN(w./o. DPMB)+M2ER is the full model withou without DPMB along with M2ER. Tab. 3 and Fig. 8 present a detailed analysis of various configurations ($\#\mathrm{Conf}_1$ to $\#\mathrm{Conf}_4$). Results show that omitting any component significantly degrades performance, underscoring the importance of each component's design and integration. Both quantitative and visual assessments reveal the critical role of component synergy, highlighting their collective contribution to optimal reconstruction performance.

**Ablation Studies on Loss Functions.** Tab. 3 presents the results from different combinations of loss function. The analysis shows that incorporating $\mathcal{L}_p$ and $\mathcal{L}_{kl}$ significantly improves reconstruction performance. Notably, PSNR and SSIM increase and then decrease as the $\lambda_1$ and $\lambda_2$. Considering the goal is to enhance perceptual quality (high LPIPS), we set $\lambda_1 = 0.01$ and $\lambda_2 = 0.1$ for training.

Table 3: Ablation study on components and loss functions

| ID | Config | PSNR (dB) ↑ | SSIM ↑ | LPIPS ↓ |
|---|---|---|---|---|
| $\#\mathrm{Conf}_1$ | TFR | 10.53 | 0.2604 | 0.5935 |
| $\#\mathrm{Conf}_2$ | TFR + MSRN | 17.29 | 0.5248 | 0.4262 |
| $\#\mathrm{Conf}_3$ | TFR + MSRN (w./o. DPMB) +$M^2$ER | 20.39 | 0.5782 | 0.3568 |
| $\#\mathrm{Conf}_4$ | Full model | 23.63 | 0.7527 | 0.2533 |
| $\#\mathrm{Conf}_5$ | $\mathcal{L}_{mse}$ | 22.67 | 0.6932 | 0.2569 |
| $\#\mathrm{Conf}_6$ | $\mathcal{L}_{mse} + 0.01 * \mathcal{L}_p$ | 22.52 | 0.7095 | 0.2527 |
| $\#\mathrm{Conf}_7$ | $\mathcal{L}_{mse} + 0.1 * \mathcal{L}_p$ | 22.61 | 0.7233 | 0.2531 |
| $\#\mathrm{Conf}_8$ | $\mathcal{L}_{mse} + 1.0 * \mathcal{L}_p$ | 22.95 | 0.7488 | 0.2534 |
| $\#\mathrm{Conf}_9$ | $\mathcal{L}_{mse} + 0.01 * \mathcal{L}_p + 0.01 * \mathcal{L}_{kl}$ | 23.29 | 0.7501 | 0.2437 |
| $\#\mathrm{Conf}_{10}$ | $\mathcal{L}_{mse} + 0.01 * \mathcal{L}_p + 0.1 * \mathcal{L}_{kl}$ | 23.63 | 0.7527 | 0.2533 |
| $\#\mathrm{Conf}_{11}$ | $\mathcal{L}_{mse} + 0.01 * \mathcal{L}_p + 1.0 * \mathcal{L}_{kl}$ | 23.41 | 0.7512 | 0.2529 |

## 5 CONCLUSION

In this paper, we frame lensless image reconstruction as a joint MAP problem, estimating both model mismatch error ($M^2E$) and thus high-resolution images. To enhance $M^2E$ estimation, we introduce an explicit latent space representation with proposed mathematical model. We then propose a multi-stage reconstruction network by unfolding the MAP estimator with a learned LSM prior and estimated $M^2E$. Both the scale prior coefficient and local means of the LSM model are learned through customized networks, with all parameters optimized end-to-end. Experiments show that our method outperforms state-of-the-art approaches. Future work will explore spatially varying PSF and broader generalization to other lensless cameras with lower complexity.

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

# A APPENDIX

## A.1 DETAILS OF LENSLESS IMAGING MODEL

**Wave-based Point Spread Function (PSF) Model.** Fig. 9 provides the wave-based lensless forward imaging model. We consider a single refractive or diffractive optical element, such as a thin phase mask. This element delays the phase of a complex-valued wave field proportionally to its pattern $h$

$$\phi(x', y') = \frac{2\pi\Delta n}{\lambda} h(x', y'), \tag{25}$$

where $(x', y')$ indicates the coordinates of the mask plane. $\phi(x', y')$ is the phase bound to the thin phase mask. $\lambda$ is the wavelength and $\Delta n$ is the refractive index difference between air ($n_{\text{air}}$) and the material of the optical mask ($n_{\text{mask}}$).

A wave field $U_\lambda$ with amplitude $A$ and phase $\phi_d$ incident on the optical mask is affected as

$$U_\lambda(x', y', z = 0) = A(x', y') e^{i(\phi_d(x',y')+\phi(x',y'))}, \tag{26}$$

where $U_\lambda(x', y', z)$ is the wave field passing through the optical element. As illustrated in Fig. 9, after the field propagates in free space at distance $z$, the field becomes

$$U_\lambda(x, y, z) = \frac{e^{ikz}}{i\lambda z} \iint U_\lambda(x', y', 0) e^{\frac{ik}{2z}\left((x-x')^2+(y-y')^2\right)} dx' dy', \tag{27}$$

which applies the Fresnel propagation operator, an accurate model for near and far distances when $\lambda \ll z$. The wavenumber is $k = 2\pi/\lambda$.

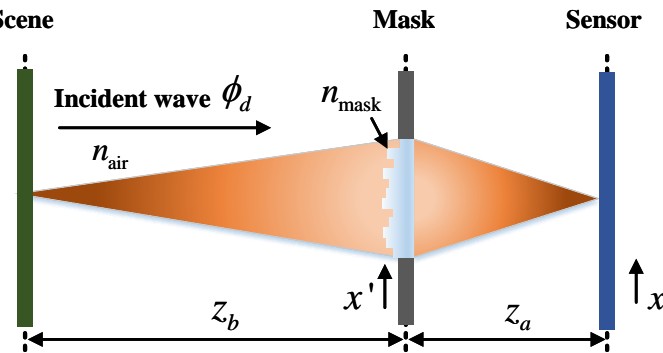

Figure 9: Wave-based lensless forward imaging model.

Let $\Phi$ be PSF associated with the optical mask, a point representing an optical infinity, the optical axis at the front of the sensor arriving at a distance $z$ from the element propagates through the element as

$$\Phi(x,y) \propto \left| \mathcal{F}\left\{ A(x',y')\, e^{i\phi(x',y')} e^{i\frac{\pi}{\lambda z}(x'^2 + y'^2)} \right\} \right|^2,  \tag{28}$$

where $\mathcal{F}\{\cdot\}$ is the Fourier transform (FT).

**Lensless Imaging Model.** Considering a single depth, $\mathbf{x}$ is the intensity of the natural object at this depth slice. According to the convolution model, the lensless imaging measurement $\mathbf{y}$ is:

$$\mathbf{y} = \Phi \circledast \mathbf{x} + \mathbf{n} = \mathcal{O}\mathbf{x} + \mathbf{n}  \tag{29}$$

where $\circledast$ represents convolution operation, $\mathbf{n}$ is the noise term. Note that we default the model $\mathcal{O}$ as the agent of PSF to unify the description. Thus, the model mismatch and PSF mismatch are equivalent.

## A.2 SYSTEM SETUPS OF OUR FINCAM

As shown in Fig. 10 (a), the FinCam we constructed consists of a phase mask, image sensor, occlusion support, and optical aperture. The pattern of phase mask is produced by SFinGe algorithm (Cappelli (2009)), resulting in a high-contrast, randomly textured fingerprint image, as shown in Fig. 10 (b). The occlusion support encloses the imaging system to block stray light from the surroundings. The optical aperture is attached to the phase mask to ensure that light enters the imaging system only through this aperture. The phase mask is positioned 2 mm in front of the image sensor and is secured with support material. Manufactured using two-photon lithography 3D printing, the phase mask measures $2.5\,\text{mm} \times 2.5\,\text{mm}$. The practical setup of FinCam is shown in Fig. 10 (c). The FinCam is equipped with a $6.41$ MP Sony IMX178 CMOS sensor with $2.4\mu\text{m} \times 2.4\mu\text{m}$ pixels and a 12-bit color depth.

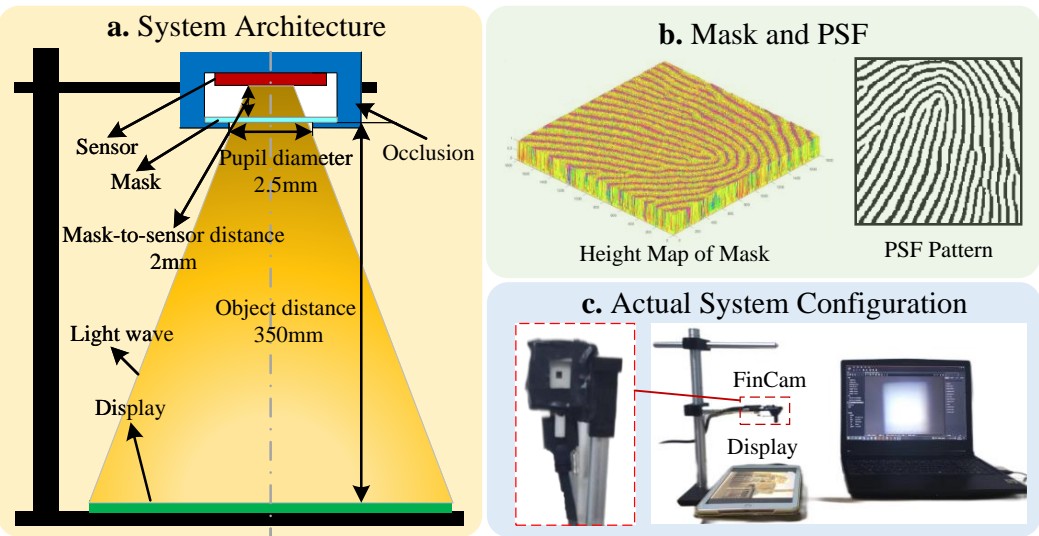

Figure 10: The hardware setup of FinCam.

We create a large dataset called DCD-FinCam by projecting images onto monitors and capturing these projections with lensless cameras. This ensures alignment with the true imaging model for lensless cameras and facilitates the collection of a labeled dataset for lensless image reconstruction. The example of the DCD-FinCam dataset is shown in Fig. 11.

## A.3 THE DETAILS OF DPMB

The DPMB is designed with an encoder-decoder architecture to utilize the multi-scale features, as shown in Fig. 12. Specifically, in the encoder, the 1-st and 2-nd scales consist of channel attention

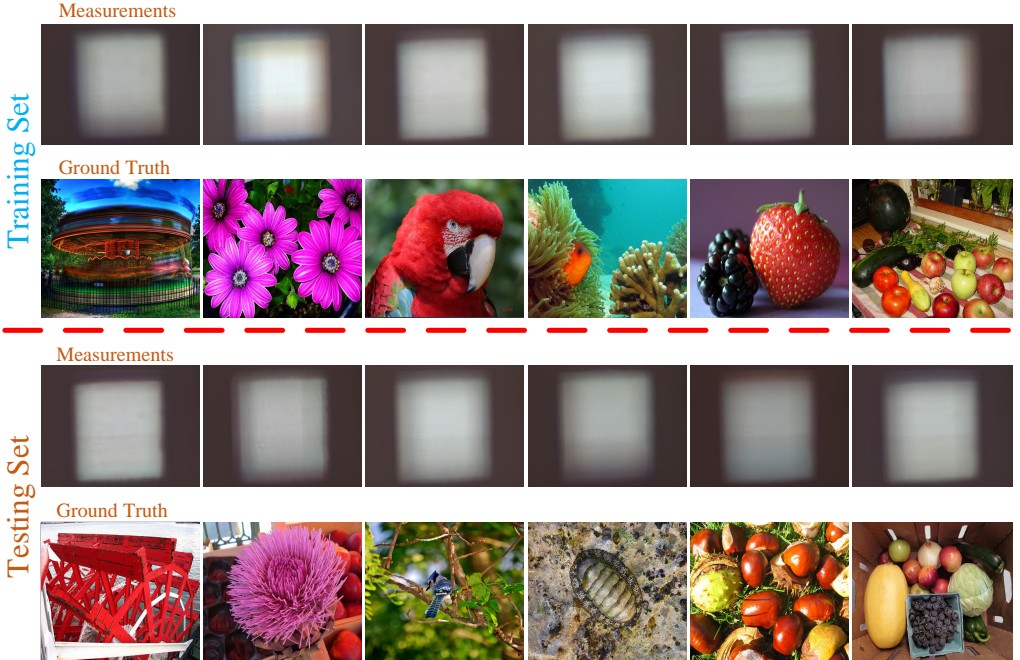

Figure 11: Examples of the DCD-FinCam dataset.

block (CAB), residual block (RB), inline feature fusion block (IFFB), and down-sampling (simplified as Down), while the 3-rd scale consists of CAB, RB, IFFB, Down, and convolution. In the decoder, the 1-st and 2-nd scales consist of up-sampling (simplified as Up), RB, and CAB, while the 3-rd scale consists of convolution, Up, RB, and CAB.

For the sake of subsequent description, the encoder and decoder features extracted from the $k$-th stage are represented as

$$
\begin{aligned}
\mathbf{F}_{\mathrm{Enc}}^{k} &= \mathrm{Cat}\left(f_{\mathrm{Enc}}^{k,1}, f_{\mathrm{Enc}}^{k,2}, f_{\mathrm{Enc}}^{k,3}\right) \\
\mathbf{F}_{\mathrm{Dec}}^{k} &= \mathrm{Cat}\left(f_{\mathrm{Dec}}^{k,1}, f_{\mathrm{Dec}}^{k,2}, f_{\mathrm{Dec}}^{k,3}\right)
\end{aligned}
\tag{30}
$$

where the $\mathrm{Cat}(\cdot)$ is the concatenation operation. The $\{f_{\mathrm{Enc}}^{k,1}, f_{\mathrm{Enc}}^{k,2}, f_{\mathrm{Enc}}^{k,3}\}$ and $\{f_{\mathrm{Dec}}^{k,1}, f_{\mathrm{Dec}}^{k,2}, f_{\mathrm{Dec}}^{k,3}\}$ are transmitted in IFFB in the encoder and RB in the decoder at different stages to integrate beneficial cues at different scales.

The CAB at each scale in the encoder and decoder is employed to enhance the representation of specific features, facilitating the capture and utilization of information conducive to reconstruction. The steps of CAB can be mathematically detailed as

$$
f_{\mathrm{AP}}^{k,i} = \mathrm{AP}\left(\mathrm{CR}\left(\mathrm{CR}\left(f^{k,i}\right)\right)\right),
\tag{31}
$$

$$
f_{\mathrm{W}}^{k,i} = \mathrm{Sigmoid}\left(\mathrm{CR}\left(\mathrm{CR}\left(f_{\mathrm{AP}}^{k,i}\right)\right)\right),
\tag{32}
$$

$$
f_{\mathrm{CA}}^{k,i} = f_{\mathrm{W}}^{k,i} \odot f_{\mathrm{AP}}^{k,i} + f^{k,i},
\tag{33}
$$

where $\mathrm{AP}(\cdot)$ and $\mathrm{Sigmoid}(\cdot)$ are the average pooling operator and sigmoid function, respectively.

The RB at each scale in the encoder and decoder are exploited to enhance the ability to capture crucial features. Mathematically, the RB in encoder is described as $f_{\mathrm{RB}}^{k,i} = \mathrm{CR}\left(\mathrm{CR}\left(f_{\mathrm{IN}}^{k,i}\right)\right) + f_{\mathrm{IN}}^{k,i}$, while the RB in decoder is $f_{\mathrm{RB}}^{k,i} = \mathrm{CR}\left(\mathrm{CR}\left(f_{\mathrm{IN}}^{k,i} + f_{\mathrm{Enc}}^{k,i}\right)\right) + f_{\mathrm{IN}}^{k,i}$.

The IFFB at each scale in the encoder fuses the inter-stage information to balance the intrinsic information loss. We compute two affine parameters $\sigma^{k,i}, \mu^{k,i} \in \mathbb{R}^{C \times H \times W}$ to transfer the intermediate

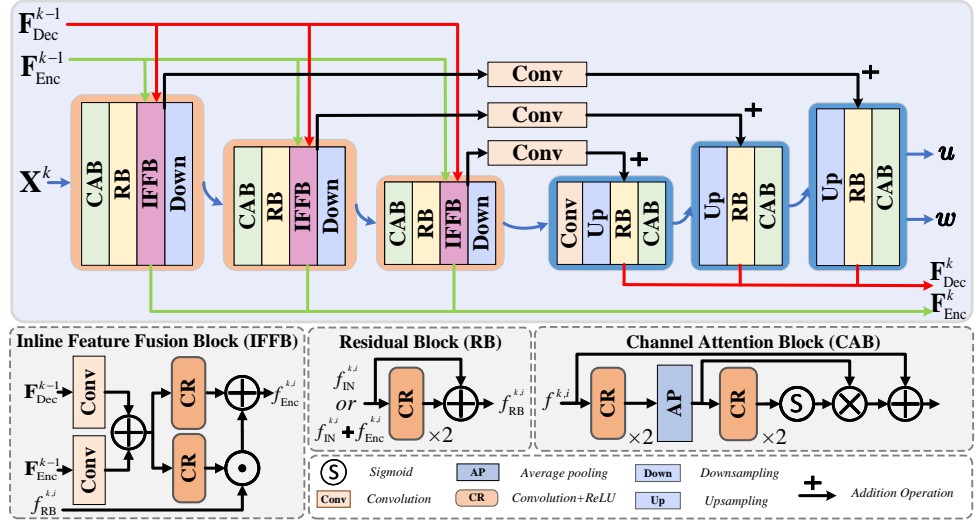

Figure 12: The architecture of DPMB.

output $f_{\mathrm{RB}}^{k,i} \in \mathbb{R}^{C \times H \times W}$ (the output of RB in encoder at $k$-th stage and $i$-th scale) to an informative one $f_{\mathrm{Enc}}^{k,i} \in \mathbb{R}^{C \times H \times W}$,

$$T_n^{k,i} = \mathrm{Conv}\left(\mathbf{F}_{\mathrm{Enc}}^{k-1}\right) + \mathrm{Conv}\left(\mathbf{F}_{\mathrm{Dec}}^{k-1}\right), \tag{34}$$

$$\sigma^{k,i} = \mathrm{CR}\left(T_n^{k,i}\right), \mu^{k,i} = \mathrm{CR}\left(T_n^{k,i}\right), \tag{35}$$

$$f_{\mathrm{Enc}}^{k,i} = f_{\mathrm{RB}}^{k,i} \odot \sigma^{k,i} + \mu^{k,i}, \tag{36}$$

where $\mathrm{Conv}(\cdot)$ is the convolution with a kernel size of $3 \times 3$.

The feature fusion described above is known as spatial-adaptive normalization. Unlike conditional normalization techniques (Ulyanov et al. (2017)), the parameters $\sigma^{k,i}, \mu^{k,i} \in \mathbb{R}^{C \times H \times W}$ are spatial tensors instead of vectors. $\sigma^{k,i}$ and $\mu^{k,i}$ enable the encoder and decoder to capture multi-scale features while retaining the refined memory from previous stages, ensuring that each scale retains well-preserved spatial information. Consequently, the resulting proximal mapping is more informative. To denote the set of multi-scale encoder and decoder features, $i.e.$, $\mathbf{F}^k = \left\{\mathbf{F}_{\mathrm{Enc}}^k, \mathbf{F}_{\mathrm{Dec}}^k\right\}$, our DPMB is expressed as

$$\hat{\mathbf{X}}^k, \mathbf{F}^k = \mathrm{DPMB}\left(\hat{\mathbf{X}}^{k-1}, \mathbf{F}^{k-1}; \boldsymbol{\theta}^k\right), \tag{37}$$

where $\boldsymbol{\theta}^k$ refers to the parameters of the DPMB at $k$-th stage.

## A.4 COMPARISONS WITH OTHER STATE-OF-THE-ARTS ON FINCAM

We present a comprehensive comparison between our $\mathrm{M}^2\mathrm{LNet}$ and other state-of-the-art methods considering the model mismatch, namely MMCN (Zeng & Lam (2021)), FlatNet (Salman et al. (2022)), MN-FISTA-Net (Qian et al. (2024)), and MWDNS (Li et al. (2023)) to meticulously evaluate their reconstruction performance on DCD-Fincam dataset captured by FinCam, as shown in Fig. 13 and Tab. 4. The comparison results shows our method maintains state-of-the-art performance in both visual quality and quantitative evaluation.

## A.5 RECONSTRUCTION RESULT FOR NATURAL SCENES

To further validate the generalization capability of our method, we collected natural scene data using a custom-built FinCam and compared it with top-performing methods, as illustrated in Fig. 14. The selected methods successfully reconstruct underlying scene information from complex lensless imaging measurements, demonstrating the effectiveness of our custom FinCam. Moreover, our

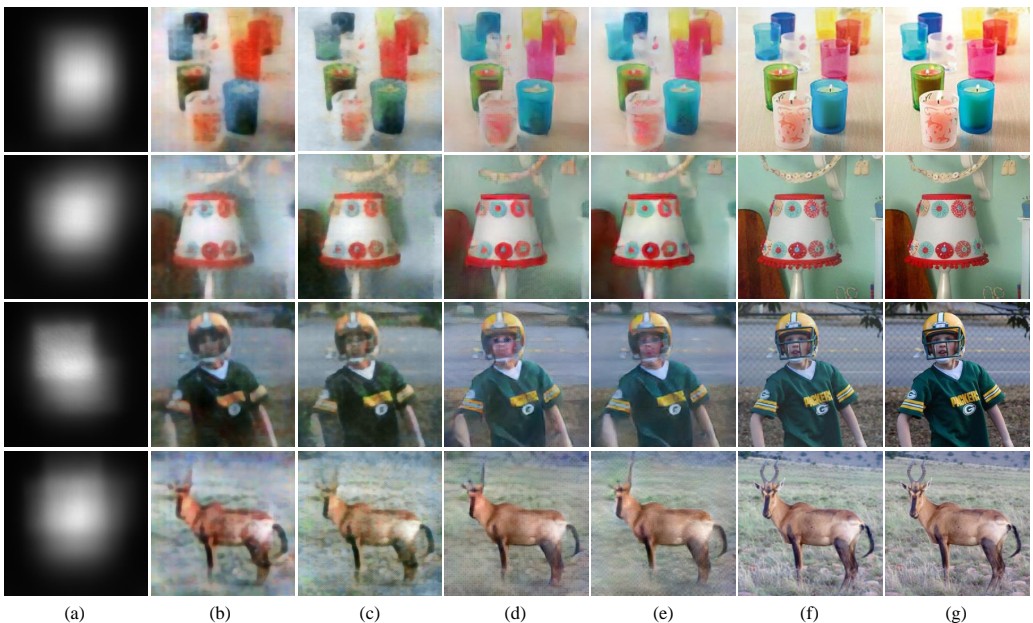

(a)      (b)      (c)      (d)      (e)      (f)      (g)

Figure 13: Visual inspection of the reconstruction performance for FinCam by (b) MMCN, (c) FlatNet, (d) MN-FISTA-Net, (e) MWDNS, and (f) our $M^2$LNet. (a) is the lensless imaging measurements corresponding to (g) ground truths.

Table 4: Comparison of reconstructed performance on DCD-FinCam. The best results are shown in red.

| Method | PSNR (dB) ↑ | SSIM ↑ | LPIPS ↓ |
|---|---|---|---|
| MMCN (Zeng & Lam (2021)) | 20.44 | 0.5487 | 0.3307 |
| FlatNet (Salman et al. (2022)) | 20.07 | 0.6017 | 0.3135 |
| MN-FISTA-Net (Qian et al. (2024)) | 21.97 | 0.5623 | 0.3117 |
| MWDNS (Li et al. (2023)) | 22.36 | 0.5779 | 0.2935 |
| $M^2$LNet (ours) | 24.19 | 0.7566 | 0.2533 |

method consistently outperforms in visual quality, further confirming its robust generalization capability. This experiment provides valuable insights for advancing the practical application of lensless imaging technology.

### A.6   ABLATION STUDIES ON STAGE NUMBER OF MSRN.

We investigated the impact of the stage number of MSRN, varying from 0 to 5. The results, shown in Figs. 15 and 16, reveal that performance improves with the addition of stages, but levels off around 4 stages. Beyond this point, further increases the number of stages do not significantly enhance performance. To balance efficiency and computational cost, we select 4 stages, optimizing performance while controlling computational burden.

### A.7   COMPARISON WITH STATE-OF-THE-ARTS ON DIFFUSERCAM.

To further validate the generalization capability of our method, we conduct experiments on the publicly available dataset provided by the DiffuserCam prototype (Monakhova et al. (2019)). Adhering to its data configuration protocols in Monakhova et al. (2019), we compare reconstruction results across methods such as MMCN (Zeng & Lam (2021)), FlatNet (Salman et al. (2022)), MWDNs (Li et al. (2023)), MDGAN (Ni et al. (2024)), and ours. The results in Fig. 17 highlight our method's ability to recover detailed scene information effectively, demonstrating its applicability to Diffusercam setups.

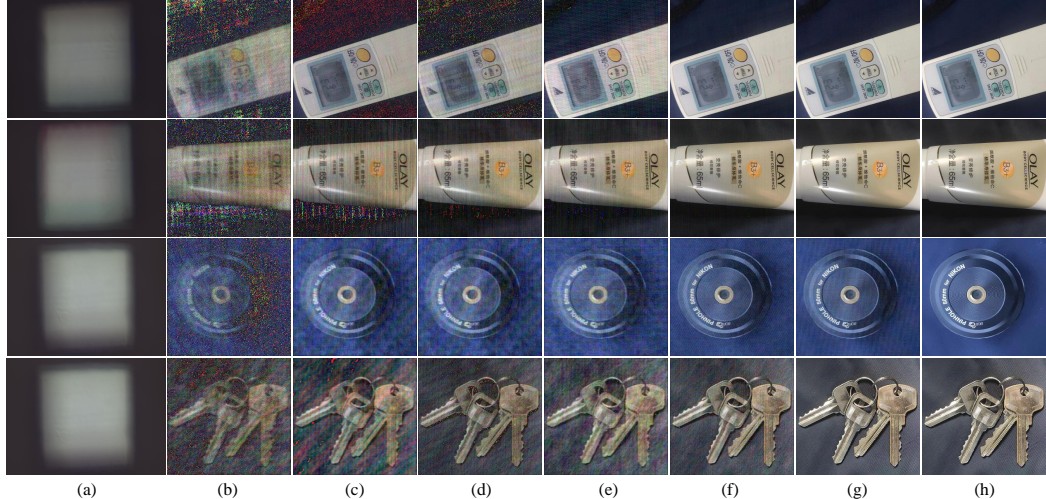

Figure 14: Visual inspection of the reconstruction performance for natural scenes captured by Fin-Cam with (b) UDN (Banerjee et al. (2023)), (c) MMCN (Zeng & Lam (2021)), (d) MN-FISTA-Net (Qian et al. (2024)), (e) MWDNS (Li et al. (2023)), (f) ULAMP-Net (Yang et al. (2022)), (g) MDGAN (Ni et al. (2024)), and (h) our $M^2$LNet. (a) is the lensless imaging measurements.

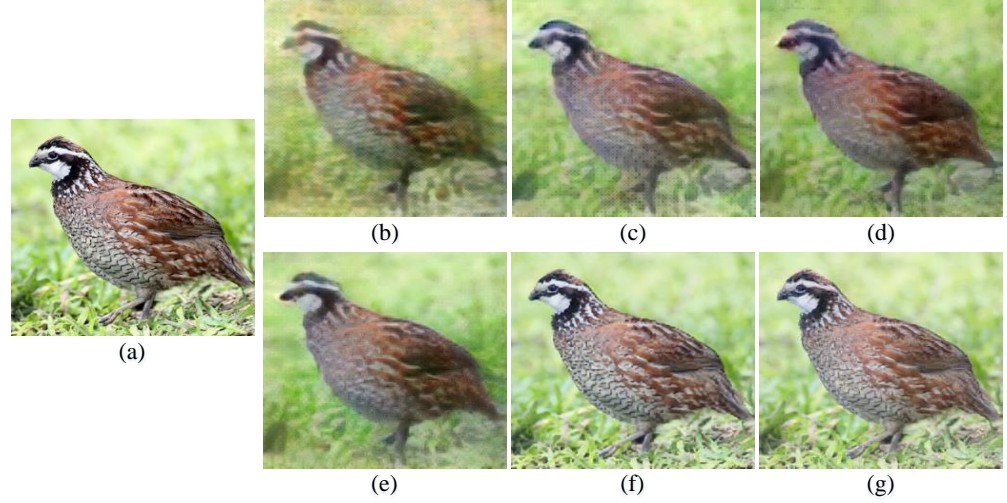

Figure 15: Visual results of ablation study on stage number of MSRN. (b)–(g) bind to the stage number from 0 to 5. (a) is the ground truth.

## A.8 COMPARISON RESULTS BY LATEST METHODS.

To further demonstrate the superiority of our method, we select the two most recent methods (DPNN and DeepLIR) for comparison experiments, with the corresponding visualization results presented in Fig. 18. As shown, our method continues to demonstrate superior performance.

## A.9 LIMITATIONS

In general, our method achieves high-precision visual reconstruction under $M^2$E. However, experiments show that it is currently effective for minor $M^2$E such as translations, rotations, and slight PSF blur. Future work will explore to enhance generalization and practicality.

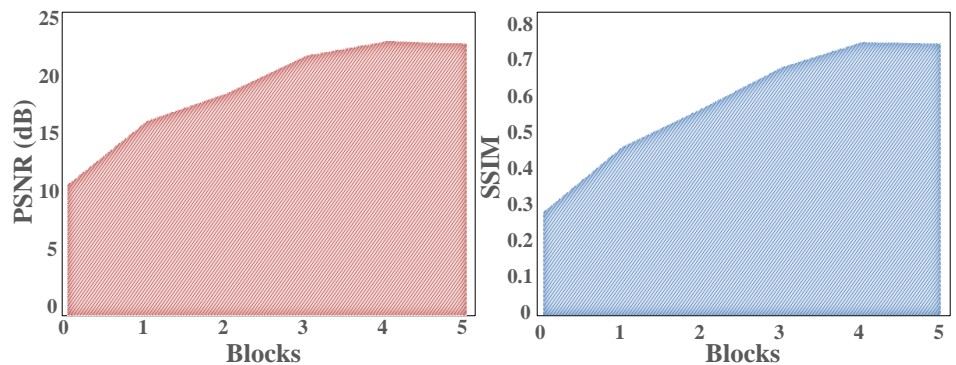

Figure 16: Quantitative ablation study on the effect of MSRN with stage number from 0 to 5.

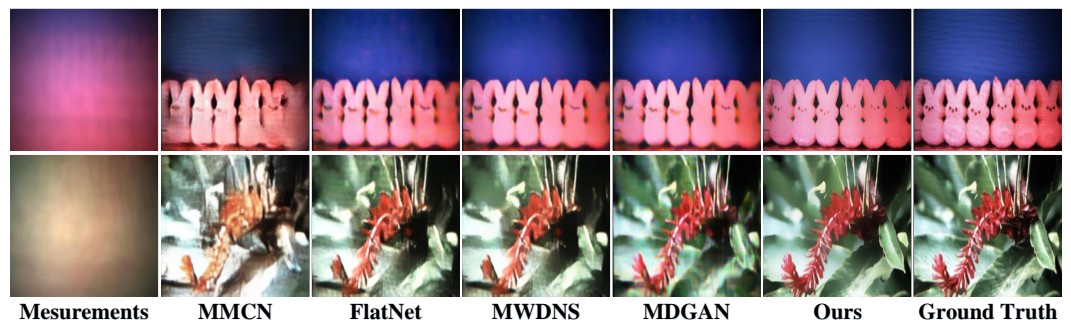

Figure 17: Visual inspection of the reconstruction performance on DiffuserCam.

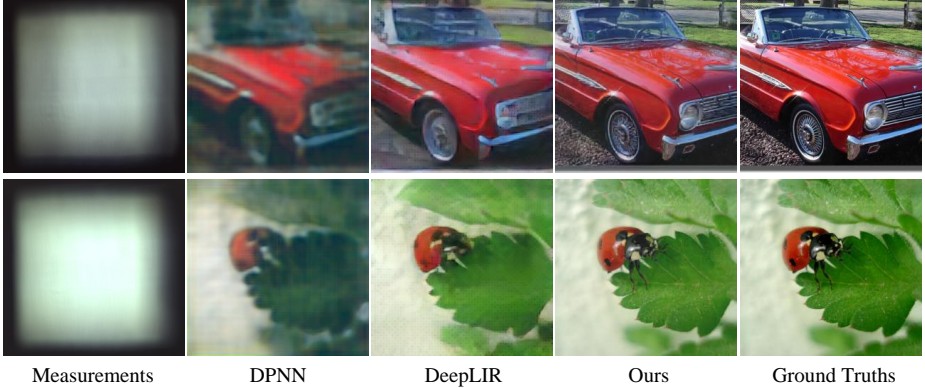

Figure 18: Visual inspection of the reconstruction performance by latest methods such as DPNN and DeepLIR.

