# OpenReview forum: "Enhancing lensless imaging via Explicit Learning of Model Mismatch"
_ICLR.cc/2025/Conference — ICLR 2025 Conference Withdrawn Submission_

### Official Review · Reviewer_ZAam · 2024-11-04

**Soundness:** 3
**Presentation:** 3
**Contribution:** 2
**Rating:** 6
**Confidence:** 3

**Summary:**

This paper focuses on  inaccuracy in lensless imaging reconstruction caused by mismatched point spread function (PSF) models. It introduces a unrolling model, using a Maximum a Posteriori (MAP) framework with a Laplacian Scale Mixture (LSM) prior. The method iteratively estimates both the image and model mismatch error (M2E) by updating reconstructions (x^{k+1}) based on prior estimates (x^k, A^k), and obtaining the parameters (A^{k+1}) through the M2ER (model mismatch error representation) model. Experimental results demonstrate that this approach improves reconstruction quality compared to state-of-the-art methods.

**Strengths:**

1. The presentation is very clean and straightforward.
2. The equations and variables are well defined, helping in understanding the methodology.
3. Combining the unrolling method with traditional statistical model-based denoising and deep learning-based denoising is a promising approach to improve reconstruction accuracy.
4. The dataset is large, and the comparison methods are well-chosen, leading reliability to the results.

**Weaknesses:**

The main concern for me is about the problem setting, especially in Eq. (5), (7), and (8). See my questions below.

**Questions:**

1. The equation (5) is equivalent to p(A, x | x^) p(x^) = p(A, x, x^) = p(x | A, x^)  p(A | x^)  p(x^). Applying the log-likelihood, we get log p(A, x | x^) + log p(x^) = log p(x | A, x^) + log p(A | x^) + log p(x^) = log p(x | A, x^) + log p(A | x^) + log p(Ax + ksi). How does this lead to equation (6)? I don’t think adding log p(x) makes it proportional to log p(A, x | x^). Could you provide a step-by-step derivation of Equation (6) from Equation (5) and explain why introducing log p(x) supports your proportionality claim?

2. From equation (7) to (8a), why is A* = arg max_A log p(A | x^) + log p(x^ | A, x) equivalent to (8a)? I think you treat the second term, log p(x^ | A, x), as a constant, but in general, it isn’t. Do you make any assumptions in this step? If so, could you justify the reasonableness of this assumption?

3. Similarly, from equation (7) to (8b), you remove the term log p(A | x^). However, log p(A | x^) = log p(A | Ax + ksi). Why is this term always considered a constant when handle x?

4. For equation (16), I think this is a discrete model for w_i. Why do you use an integral here? Is this intended to represent a continuous approximation of the discrete model?

5. I am also curious how much adding L_p in the loss function can improve accuracy.

6. Will your model extend to Poisson or negative binomial noise for low-light images?

---

> ### Author Response · Authors · 2024-11-25
>
> We deeply appreciate your insightful comments and recognition of our work, and have provided detailed responses to each point to ensure clarity and address your concerns comprehensively.
>
> **Questions:**
>
> **Q1:About the derivation from Eq. (5)  to  (6)**. We appreciate your detailed feedback about Eqs. (5) and (6). Below is the step-by-step derivation from Eq. (5) to Eq. (6). Restating Eq. (5):
>
> $$
> p(A,x|\hat{x}) = p(A|\hat{x}) p(x|A,\hat{x}),
> $$
>
> we apply Bayes’ rule to $p(x|A,\hat{x})$:
>
> $$
> p(x|A,\hat{x}) = \frac{p(\hat{x}|A,x)p(A|x) p(x)}{p(\hat{x}|A)p(A)}.
> $$
>
> Since $x$ and $A$ are independent, we have $p(A|x)=p(A)$, which implies that
>
> $$
> p(x|A,\hat{x}) = \frac{p(\hat{x}|A,x) p(x)}{p(\hat{x}|A)}.
> $$
>
> Substituting this back into Eq. (5), we get:
>
> $$
> p(A,x|\hat{x})=p(A|\hat{x}) \cdot \frac{p(\hat{x}|A,x) p(x)}{p(\hat{x}|A)}.
> $$
>
> Here, $p(\hat{x}|A)=\int p(\hat{x}|A,x)p(x)dx$ serves as a normalization constant, ensuring the conditional probability is normalized. It does not affect the optimization of $\log p(A,x|\hat{x})$ and can therefore be omitted when focusing on proportionality. Simplifying, we express Eq. (5) as:
>
> $$
> p(A,x|\hat{x}) \propto p(A|\hat{x}) p(\hat{x}|A,x) p(x).
> $$
>
> Finally, taking the logarithm of both sides leads directly to Eq. (6).
>
> **Q2:the derivation from Eq. (7) to Eq. (8a).** We sincerely appreciate your thoughtful question regarding the derivation from Eq. (7) to Eq. (8a). In Eq. (7), we have the joint log-likelihood:
>
> $$
> (A^\*, x^\*) = \arg \max_{A,x} \log p(A|\hat{x}) + \log p(\hat{x}|A,x) + \log p(x).
> $$
>
> To optimize it efficiently, we adopt a decomposition strategy, as outlined in Sec. 3.2 of reference [1], splitting Eq. (7) into two subproblems (Eq. (8a) and Eq. (8b)) for sequential optimization. The transition to Eq. (8a) involves maximizing the log-likelihood over A while treating $x$ as fixed. Specifically, the terms $\log p(\hat{x}|A, x) $ and $\log p(x)$ are treated as constants with respect to $A$ during this step, simplifying the optimization to: $A^\* = \arg \max_A \log p(A|\hat{x})$.
>
> This assumption is justified because, at this stage, $x$ is not a variable in the optimization over $A$; instead, it serves as a fixed input derived from the previous iteration. Such a decomposition aligns with similar approaches in joint optimization problems, enabling tractable solutions while preserving the overall consistency of the framework.
>
> [1] Zhao Q, Wang H, Yue Z, Meng D. A deep variational Bayesian framework for blind image deblurring. Knowledge-Based Systems. 5, 2022, 249:109008.
>
> **Q3:the derivation from Eq. (7) to Eq. (8b).** We sincerely appreciate your insightful question regarding the derivation from Eq. (7) to Eq. (8b). In this step, the goal is to optimize for $x$ under the assumption that A is fixed. This treatment is consistent with the decomposition strategy detailed in Sec. 3.2 of reference [1], which employs conditional optimization for tractability. The original optimization problem in Eq. (7) is:
>
> $$
> (A^\*, x^\*) = \arg \max_{A,x} \log p(A|\hat{x}) + \log p(\hat{x}|A,x) + \log p(x).
> $$
>
> Combining with statement in Q2, when optimizing with respect to $x$, the term $ \log p(A|\hat{x})$ becomes independent of $x$ because it depends only on $A$ and $\hat{x}$. Since $A$ is fixed in this step, $\log p(A|\hat{x})$ does not affect the optimization for $x$ and can be treated as a constant. This simplification reduces the problem to:
>
> $$
> x^\* = \arg \max_x \log p(\hat{x}|A,x) + \log p(x).
> $$
>
> This decomposition allows us to decouple the joint optimization problem into two manageable subproblems: Eq. (8a) for $A$ and Eq. (8b) for $x$. By isolating these processes, we ensure more efficient optimization and targeted parameter learning for each variable. This stepwise method effectively balances model simplicity and performance, enabling robust parameter estimation.
>
> **Q4: About the Eq. (16).** We appreciate your detailed feedback about the Eq. (16). In Eq. (16), the integral is used to represent a continuous approximation of the discrete model for $w_i$. While $w_i$ is inherently discrete for each pixel, the integral is applied to model the prior distribution of $w_i$ as a continuous random variable. This is a standard method to incorporate uncertainty in $w_i$, allowing for a more flexible and tractable representation of the prior distribution. The use of the integral instead of a discrete sum simplifies the optimization and captures the variability of $w_i$ efficiently, making it a reasonable approximation for modeling continuous scale parameters.

---

> > ### Author Response · Authors · 2024-11-25
> >
> > **Q5: how much adding Lp in the loss function can improve accuracy?** Thank you for your question. We investigated the impact of adding L_p to the loss function through experiments, as shown in  **Table 3** . Without Lp, the metrics are: PSNR=22.67 dB (higher is better), SSIM=0.6932 (higher is better), and LPIPS=0.2569 (lower is better). When Lp was added with a weight of 0.01 (i.e., 0.01×Lp), we observed slight improvements: SSIM increased to 0.7095, and LPIPS decreased to 0.2527, though there was a minor drop in PSNR. Increasing the weight of Lp to 0.1 (i.e., 0.1×Lp) results in further improvements with SSIM reaching 0.7233 and LPIPS at 0.2531, while PSNR decreased slightly. Finally, when Lp was set to a weight of 1.0 (i.e., 1.0×Lp), we observed notable improvements: PSNR increased to 22.95, SSIM rose to 0.7488, and LPIPS decreased to 0.2534. This corresponds to a 1.24% improvement in PSNR, an 8.02% improvement in SSIM, and a 1.36% improvement in LPIPS compared to the baseline without Lp.
> >
> > These results show that adding Lp improves perceptual quality, particularly with higher weights. The increase in SSIM and decrease in LPIPS reflect better alignment with human visual perception, as these metrics more effectively capture perceptual quality compared to pixel-wise metrics like PSNR. Overall, while the exact impact of adding Lp may vary depending on the specific task and dataset, our experiments consistently show that adding Lp leads to improved perceptual performance, making the model output more visually pleasing and closer to human judgment of image quality.
> >
> > **Q6: Will your model extend to Poisson or negative binomial noise for low-light images?** Thank you for your question. Our method is designed primarily for natural light scenarios and is not directly applicable to low-light image processing tasks. However, extending the paradigm to such cases is feasible with modifications. Specifically, the likelihood function would need to be adapted to incorporate noise models relevant to low-light conditions. For instance, Poisson noise can be modeled using a Poisson likelihood function, while negative binomial noise, which accounts for overdispersion, requires a more flexible distribution. Additionally, integrating a low-light inversion model into our lensless imaging framework would be necessary to reshape the network for these scenarios. This extension would allow the model to more effectively handle the unique characteristics of low-light imaging scenarios.

---

> > ### Comment · Reviewer_ZAam · 2024-11-25
> > **Good!**
> >
> > good

---

> > > ### Author Response · Authors · 2024-12-03
> > >
> > > Thanks for your thoughtful feedback. We sincerely hope that the revisions made have sufficiently addressed your concerns. If possible, we would greatly appreciate it if you could kindly reconsider the score in light of these improvements.

---

### Official Review · Reviewer_Vo5z · 2024-11-05

**Soundness:** 3
**Presentation:** 4
**Contribution:** 3
**Rating:** 6
**Confidence:** 4

**Summary:**

This paper addresses the challenge of model mismatch error (M2E) in lensless imaging, which occurs due to inaccuracies in the point spread function (PSF) model. The authors propose a novel method called M2LNet that simultaneously estimates M2E and reconstructs high-resolution images from lensless measurements. The contributions of this work including: 1) Joint MAP Estimation: Formulate lensless image reconstruction as a joint Maximum a Posteriori (MAP) problem to co-estimate M2E and reconstruct the underlying scene. 2) M2E Representation (M2ER): Introduces an explicit learning model for M2E, improving robustness against PSF inaccuracies by learning both the feature (mean) and uncertainty (variance) in the latent space. 3) Multi-Stage Reconstruction Network (MSRN): Develops a multi-stage network by unfolding the MAP estimator with a learned Laplacian Scale Mixture (LSM) prior and M2E representation (M2ER), optimizing all parameters end-to-end. The extensive experiments on datasets captured by PHlatCam and FinCam prototypes demonstrate that M2LNet significantly surpasses some existing methods in terms of image quality and reconstruction accuracy.

**Strengths:**

1. The explicit learning of M2E with a dedicated MAP module is a novel approach that addresses the limitations of previous methods, which either ignored M2E or treated it as a noise correction step.
2. The paper builds upon a solid theoretical framework, providing a clear mathematical formulation of the lensless imaging problem and the proposed solutions.
3. The paper is well-structured and organized, presenting the problem, methodology, experiments, and results in a clear and logical manner.

**Weaknesses:**

1. It is good to focus on the M2E, but I think the design of the M2ER is simple.
2. While the paper compares M2LNet with some existing state-of-the-art methods, the comparison is not sufficient, some recent works are not presented and compared in this work.
3. Some of the ablation study of the work is confusing.

**Questions:**

1. It is good to focus on the M2E in lensless imaging, but I think the design of the reconstruction model is simple.
2. what is the differences of ‘TFR + MSRN + M2ER’ and ‘Full model’ in your ablation study in Table. 3? There is a significant improvement in the reconstruction results in both PSNR and SSIM, which is not presented clearly.
3.In the ablation study on loss function in Table.3, the ‘Lmse + 1.0 * Lp’ presents the best performance, why do you choose to use ‘0.01 * Lp ’in the following tests.
4. The color highlight in Table.1 is confusing.
5.Some most recent related works are not mentioned and compared in the paper, like the following References, it would be better to compare all the SOTA works to demonstrate the performance of the proposed method.

R1: Lensless Imaging Based on Dual-Input Physics-Driven Neural Network[J], Advanced Photonics Research, 5, 11, 2024.
R2: DeepLIR: Attention-Based Approach for Mask-Based Lensless Image Reconstruction[C], Proceedings of the IEEE/CVF Winter Conference on Applications of Computer Vision. 2024: 431-439.

---

> ### Author Response · Authors · 2024-11-25
>
> We sincerely appreciate your valuable comments and recognition, and have provided detailed responses to address each point clearly and comprehensively.
>
> **Weaknesses:**
>
> **Q1:It is good to focus on the M2E, but I think the design of the M2ER is simple.** Thank you for your insightful comment. The design of $M^2ER$ is intentionally kept simple, aligning with the structure of Eq. (13) to ensure mathematical consistency as described in Eq. (4). We chose a simple Gaussian distribution to model ($\Delta_{\hat{O}}$) because it effectively captures the randomness and uncertainty caused by misalignment, OSD variations, and system noise. This choice allows for efficient modeling of both the feature (mean) and uncertainty (variance) in the latent space. While the design is simple, it strikes a balance between robustness in $M^2E$ estimation and practical efficiency, making it scalable and effective for real-world applications.
>
> **Q2:Comparison with some recent works.** Thank you for your valuable feedback. Following your advice, we have added more SOTA methods (DPNN (2024) and DeepLIR (2024)) into our evaluation. The comparative results are now presented in **Appendix A.8 and Fig. 18** of revised manuscript. As illustrated, our method continues to demonstrate superior performance.
>
> **Q3:About the ablation study.** We appreciate your comments regarding the ablation study. To clarify, the ablation study is organized as follows:
>
> **(1) Component Ablation:**
>
> - **TFR:** This is the baseline model, where only the TFR module is applied, without any additional components.
> - **TFR+MSRN:** In this step, we introduce the MSRN to the TFR model, enhancing feature extraction.
> - **TFR+MSRN(w./o. DPMB)+**$M^2ER$:** Here, we evaluate the effect of adding MSRN (without DPMB) along with $M^2ER$.
> - **Full Model:** The full model integrates all component-TFR, MSRN, and $M^2ER$-representing the complete $M^2LNet$ framework.
>
> **(2) Loss Function Ablation:** We investigate the impact of varying coefficients for the Lp and Lkl loss terms to understand their relative contributions to the overall model performance.
>
> To prevent any confusion, we have revised the corresponding description accordingly in the revised manuscript.
>
> **Questions:**
>
> **Q1:1. It is good to focus on the M2E in lensless imaging, but I think the design of the reconstruction model is simple.** Thanks for your insightful comment. The design of the reconstruction model is based on a joint MAP estimation framework, where both the $M^2E$ and the underlying scene are co-estimated. This is incorporated into the $M^2LNet $, which integrates a learned LSM prior and the estimated $M^2E$ within a multi-stage reconstruction architecture. While the model may appear simple, its design is deliberate, ensuring mathematical consistency and computational efficiency. By focusing on the joint estimation of $M^2E$ and scene reconstruction, the model effectively leverages both learned priors and $M^2E$ information to enhance reconstruction accuracy without unnecessary complexity. This design strikes a balance between performance and computational feasibility, making it suitable for real-world lensless imaging tasks.
>
> **Q2:what is the differences of ‘TFR + MSRN + M2ER’ and ‘Full model’ in your ablation study in Table. 3?** We greatly appreciate your concern regarding our ablation experiments. The difference between "TFR + MSRN (w./o. DPMB) + $M^2ER$" and the "Full model" in our ablation study is that "Full model" includes the DPMB, while "TFR + MSRN (w./o. DPMB) + $M^2ER$" excludes it. "w./o. DPMB" stands for "without DPMB". We agree that the improvement from DPMB should have been more explicitly highlighted and we have revised the manuscript (Sec. 4.6) to clarify the impact of this module.
>
> **Q3:Why do you choose to use ‘0.01 * Lp ’in the following tests?** The choice of "0.01\*Lp" in the tests was based on its lowest LPIPS value, which indicates better alignment with human visual perception. While "1.0\*Lp" achieves a slight improvements in PSNR and SSIM, it results in a higher LPIPS, indicating poorer visual quality compared to "0.01\*Lp". Considering the primary goal of reconstruction is to enhance perceptual quality, we opted for "0.01\*Lp". We apologize for the lack of explanation in the manuscript and have clarified this choice in **Sec. 4.6** of the revised manuscript.

---

> ### Author Response · Authors · 2024-11-25
>
> **Q4:The color highlight in Table.1 is confusing.** We sincerely appreciate your attention to the details in our Table 1. As noted in the caption, "The best three results are shown in red, green, and blue," indicating that the first, second, and third best performances are highlighted in red, green, and blue, respectively. We apologize for not clarifying this more explicitly in the manuscript, which may have caused confusion. We have updated the text to include this explanation.
>
> **Q5:Some most recent related works are not mentioned and compared.** Thank you for recommending additional comparison methods. Following your advice, we have incorporated the methods you suggested ([1], [2]) into our evaluation, and the comparative results are now presented in the in **Appendix A.8 and Fig. 18** of revised manuscript. These results demonstrate that our method outperforms these state-of-the-art methods. Additionally, we had already included evaluations of recent algorithms, such as MWDNS and MM-FISTA-Net, in **Appendix A.4 (Page 16) and Fig. 13 (Page 17)**, where our method also demonstrates superior performance.
>
> Reference:
> [1]Lensless Imaging Based on Dual-Input Physics-Driven Neural Network[J], Advanced Photonics Research, 5, 11, 2024.
> [2]DeepLIR: Attention-Based method for Mask-Based Lensless Image Reconstruction[C], Proceedings of the IEEE/CVF Winter Conference on Applications of Computer Vision. 2024: 431-439.

---

### Official Review · Reviewer_RFP7 · 2024-11-05

**Soundness:** 3
**Presentation:** 3
**Contribution:** 2
**Rating:** 5
**Confidence:** 3

**Summary:**

This paper addresses challenges in lensless imaging, where model mismatches, such as inaccuracies in the point spread function (PSF), impair the effectiveness of image reconstruction. The authors propose a Maximum a Posteriori (MAP) framework to jointly estimate model mismatch error (M2E) and improve image resolution. They introduce a multi-stage reconstruction network called M2LNet, which explicitly learns M2E through a gaussian latent space representation to enhance robustness against PSF inaccuracies. M2LNet, utilizing a Laplacian Scale Mixture prior, demonstrates improvements over current methods across two datasets. Extensive experiments validate its efficacy, showcasing superior image quality and computational efficiency.

**Strengths:**

1, The paper takes a unique approach by directly incorporating an explicit latent space representation of M2E within the reconstruction framework. By explicitly quantifying and integrating M2E into the multi-stage reconstruction process, the model improves robustness against inaccuracies in the point spread function (PSF), which is often a challenge in lensless imaging.

2, The proposed M2LNet framework is designed as a multi-stage reconstruction network, where each stage is unfolded from a MAP estimator and optimized in an end-to-end fashion, reducing manual finetuning burden.

3, The proposed method outperforms many existing baseline methods on two dataset, showing the effectiveness of this proposal.

**Weaknesses:**

1, The technical contribution to the broader field of computational imaging appears limited or, at the very least, unclear in the current form of this paper. Joint forward-model calibration and image reconstruction through unfolding algorithms are already well-explored in inverse imaging problems. The distinction between this approach and other end-to-end deep unfolding methods for lens design—where alternative estimation of PSF and image reconstruction are similarly implemented—remains unclear.

2, The proposed M2LNet, along with the baseline methods, is evaluated on the authors' proprietary datasets rather than on publicly available datasets, making it difficult to assess the fairness and generalizability of the results.

**Questions:**

1, The proposed method relies on two assumptions: that the M2E follows a Gaussian distribution, and that the distribution of pixel values in the underlying scene follows a Laplacian distribution. Providing a more detailed theoretical justification for these assumptions would be beneficial, especially for readers from different fields who may not be familiar with these choices.

---

> ### Author Response · Authors · 2024-11-25
>
> We sincerely appreciate your valuable comments and recognition, and have provided detailed responses to address each point clearly and comprehensively.
>
> **Weaknesses:**
>
> **Q1:About Contribution, Methodological Distinctiveness and Potential Applications.** Thanks for your insightful comment. We provide a summary of our reply in points:
>
> **(1)Broader Contributions to Computational Imaging:**
>
> - **Learnable Forward Model Estimation:** Unlike conventional deep unfolding methods, our framework introduces a model mismatch error $(M^2E)$ estimation mechanism for PSF modeling. This innovation addresses real-world challenges, such as misalignment, lateral shifts, and object-to-sensor distance variations. The method is generalizable and applicable to imaging systems with dynamic or imprecise hardware parameters, including computational holography and imaging through scattering media.
>
> - **Generalizable Framework for Inverse Problems:** The proposed framework incorporates a joint MAP estimator with a learned Laplacian Scale Mixture (LSM) prior. This methodology extends beyond lensless imaging and can be applied to other inverse problems in computational imaging, such as compressed sensing and phase retrieval. This contributes to the broader goal of enhancing the efficiency and quality of reconstruction across diverse imaging modalities.
>
> - **Application to Minimal Hardware Systems:** Lensless imaging exemplifies the potential for computational methods to replace or augment physical hardware. Our method highlights how advanced computational techniques can reduce dependence on costly or complex optical setups. This principle aligns with the broader goals of computational imaging, emphasizing efficiency and accessibility.
>
> **(2)Distinction from Existing Deep Unfolding Methods:** While joint forward-model calibration and image reconstruction using deep unfolding networks (DUNs) have been explored, our method stands out by directly addressing the “model mismatch error $(M^2E)$ ”, a crucial factor often overlooked in existing lensless imaging methodes. Conventional DUNs typically rely on approximated point spread function (PSF), which result in reconstruction inaccuracies. In contrast, we introduce a novel "latent space representation" for $(M^2E)$, explicitly modeling and correcting mismatches, improving robustness against PSF errors.
>
> Additionally, our method integrates a "Laplacian Scale Mixture (LSM) prior" in a joint MAP framework, enabling more precise and adaptive reconstruction compared to traditional PSF estimation techniques. This enhances reconstruction quality and generalization across varying imaging conditions, addressing limitations in current DUNs for lensless imaging.
>
> Our main contribution lies in the explicit modeling of $(M^2E)$ and the use of learned priors, leading to significant improvements in lensless imaging. We have revised the manuscript to better highlight these distinctions and the unique advantages of our method (in  **Related Works** ).
>
> **Q2: About Dataset and Fairness Assessment** Thanks for your comment. We would like to clarify that the DCD-PHlatCam dataset, which was used to evaluate our $(M^2LNet)$, is in fact a publicly available dataset, as described in [1]. It is widely accessible for independent verification and comparison with other methods tested on public datasets. Therefore, the evaluation results on the DCD-PHlatCam can be considered fair and generalizable in the broader context. While the DCD-FinCam dataset is proprietary, it was designed to capture more diverse and complex imaging conditions, showcasing the robustness of our method in real-world scenarios. Nonetheless, we recognize the importance of benchmarking with publicly available datasets, therefore we have revised to highlight this point more clearly in **Sec. 4.1** of revised manuscript, ensuring that the use of the DCD-PHlatCam dataset is emphasized.
>
> Reference:
>
> [1]Khan Salman, Siddique, Sundar Varun, Boominathan Vivek, Veeraraghavan Ashok, and Mitra Kaushik. FlatNet: Towards photorealistic scene reconstruction from lensless measurements. IEEE Transactions on Pattern Analysis and Machine Intelligence, 44(4):1934-1948, Oct 2022.

---

> > ### Author Response · Authors · 2024-11-25
> >
> > **Questions:**
> > **Q1:About the two assumptions: that the M2E follows a Gaussian distribution, and that the distribution of pixel values in the underlying scene follows a Laplacian distribution.**
> >
> > - **About the Gaussian distribution of $M^2E$**: Gaussian distribution is used to generate $M^2E$ because it effectively models the randomness and uncertainty introduced by misalignment, OSD variations, and system noise. Its mathematical simplicity facilitates integration into optimization frameworks, and its flexibility allows realistic simulation of PSF variations, improving reconstruction robustness.
> > - **About the Laplacian distribution** : It is used as a prior regularization term, mainly due to its ability to model the sparsity and edge characteristics inherent in natural images. Its sharp peak and heavy tails represent pixel value distributions effectively, enforce sparsity constraints, preserve edge details, and enhance reconstruction. Additionally, its flexible parameters adapt to intensity variations, improving the model's expressiveness.
> >
> > We have revised the manuscript (**Sec. 3.1** and **Sec. 3.2**) to include a more detailed explanation of these assumptions and cite relevant theoretical work to make the rationale clearer and more accessible to a broader audience.

---

### Official Review · Reviewer_vHqD · 2024-11-07

**Soundness:** 2
**Presentation:** 2
**Contribution:** 3
**Rating:** 6
**Confidence:** 4

**Summary:**

Proposes a two-step learning-based algorithms for reconstructing lensless images. The approach first estimates the forward model (or more accurately the bias introduced by inverting the measurement forward model). It then uses this forwad model estimate within an interative reconstruction network. The proposed method is tested on existing methods as well as a new dataset captured by the authors using their own lensless camera. The proposed method noticeably outperforms existing methods.

**Strengths:**

---Outperforms existing methods on experimentally captured datasets
---Introduces a new dataset
---Method is novel (to my knowledge)

**Weaknesses:**

The proposed method was trained and tested on a subset of the phlatcam dataset (images presented on a display) where there is relatively little model mismatch. The new dataset was also restricted 2D images presented on a display at a fixed distance from the sensor. There is no evidence the proposed method generalizes to real-world 3D scenes.

Many (generally minor) mathematical errors:
---"Biased" seems an inappropriate term for \hat{O}. That estimate of O may be unbiased, it's just noisy. Similarly, is fig 7's PSF "biased" or just measured.
---The text says (5) to (6) is just taking a logarithm. This step also required Bayes' rule. It's also missing a constant.
---In general, a joint maximum a posteriori estimation problem (7) (bivariate optimization) cannot be split into two independent optimization problems (8a,8b)
---Equation (8b) doesn't make any sense. You have two unknowns, \mathcal{A} and x, but you are only optimizing for one of them. Should \mathcal{A} be \mathcal{A}^*?

Presentation:
---Line 318 would be more recognizable as FFT based deconvolution by moving the inverse inside the parenthesis.
---The paper seems to be citing the latest paper to mention a result, rather than the paper where it was first introduced. For instance, it cites a 2024 paper to describe a convolutional model for a lensless camera.
---What does it mean to "mine" a variable?

**Questions:**

How well does the proposed method work on images of 3D scenes? How well does the proposed method work on the webcam captures and other real-world scenes found in the phlatcam and flatnet papers?

Would the proposed method work with diffusercams?

---

> ### Author Response · Authors · 2024-11-25
>
> Thank you sincerely for these valuable comments and acknowledgment of our work. To ensure clarity and address your concerns comprehensively, we respond to each comment individually.
> **Weaknesses**:
>
> **Q1:  There is no evidence the proposed method generalizes to real-world 3D scenes.**
>
> We appreciate this comment regarding 3D natural scene reconstruction. In **Appendix A.5 (Page 16)** and **Fig. 14 (Page 18)**, we present the reconstruction results for objects in real-world natural scenes using our FinCam lensless imaging system. The provided results demonstrate that our method generalizes effectively to natural 3D environments.
>
> **Q2: Many (generally minor) mathematical errors.**
>
> We appreciate your detailed feedback about the mathematical issues. Here is our response in points:
>
> - **About biased PSF**. In real-world systems, the measurement inaccuracies of the PSF arise not only from noise but also from various physical factors such as misalignment, lateral shift, and variations in object-to-sensor distance (OSD), all of which manifest to varying degrees. Hence, we cautiously refer to the PSF affected by these factors as the "biased PSF". Furthermore, regarding Fig. 7, it is important to note that the PSF shown in the figure is based on a simulated experiment. In real-world scenarios, distinguishing between a "biased PSF" and the "true PSF" is inherently challenging due to the complexities of measurement and system dynamics. As such, the term "biased PSF" in this context refers to the simulated measurement of the PSF, which takes into account various imperfections that would typically affect a real-world system.
> - **The process of deriving from Eq. (5) to Eq. (6)**. Indeed, the process follows Bayes' rule, and there is a normalization constant that can be ignored during optimization. To clarify the derivation from Eq. (5) to Eq. (6), we first apply Bayes' rule:
>
>   $$
>   p(x|A,\hat{x}) = \frac{p(\hat{x}|A,x) p(x)}{p(\hat{x}|A)}.
>   $$
>
>   Substituting this into Eq. (5) results in:
>
>   $$
>   p(A,x| \hat{x}) = p(A|\hat{x}) \cdot \frac{p(\hat{x}|A,x)p(x)}{p(\hat{x}|A)},
>   $$
>
>   where
>
>   $$
>   p(\hat{x}|A)=\int p(\hat{x}|A,x)p(x)dx
>   $$
>
>   is a normalization constant, which ensures the proper normalization of the conditional probability. However, since this constant does not affect the relative proportionality of the joint distribution of $A$ and $x$, it can be safely ignored when optimizing $\log p(A, x|\hat{x})$. Therefore, Eq. (5) can be further expressed as:
>
>   $$
>   p(A, x|\hat{x}) \propto p(A|\hat{x}) p(x|A, \hat{x}) p(x).
>   $$
>
>   Taking the logarithm of both sides of the Eq. (5) yields Eq. (6). We have updated the manuscript to clarify this process and mention the normalization constant.
> - **About Eq. (8a) and Eq.(8b)**. In Eq. (7), we have the joint log-likelihood:
>
>   $$
>   (A*, x*) = \arg \max_{A,x} \log p(A|\hat{x}) + \log p(\hat{x}|A,x) + \log p(x).
>   $$
>
>   To optimize this efficiently, we decompose the problem into two subproblems (Eq. (8a) and Eq. (8b)) by applying a decomposition strategy, as discussed in Sec. 3.2 of reference [1]. The transition from Eq. (7) to Eq. (8a) involves maximizing the log-likelihood over $A$, while treating terms involving $x$ as constants. Specifically, we consider $\log p(\hat{x}|A, x)$ and $\log p(x)$ as constants with respect to A because we are not optimizing over $x$ in this step. In this context, $x$ is treated as fixed, simplifying the optimization for $A$ to:
>
>   $$
>   A^\* = \arg \max_A \log p(A|\hat{x}).
>   $$
>
>   In the transition from Eq. (7) to Eq. (8b), we focus on solving $x$ under the given $A$. Since the optimization in Eq.(7) is a joint maximization problem, the term $\log p(A|\hat{x})$ becomes independent of $x$ during this step because it depends only on $A$ with the given $\hat{x}$. Since $A$ is fixed in this step, $\log p(A|\hat{x})$ does not affect the optimization for $x$ and is thus treated as a constant. Therefore, the optimization for $x$ simplifies to:
>
>   $$
>   x* = \arg \max_x \log p(\hat{x}|A,x) + \log p(x).
>   $$
> - **About the Eq. (8b) and {A\*,  x\*}**. As outlined above, Eq. (8b) focuses on solving for x under the assumption that $A$ is given. This treatment aligns with the methodology detailed in Sec. 3.2 of reference [1], where a similar conditional optimization method is employed. The decomposition of Eq. (7) into Eqs. (8a) and (8b) serves a clear purpose: to decouple the optimization of the two variables, $x$ and $A$, which are relatively independent. By isolating their optimization processes, this formulation ensures more efficient and targeted parameter learning for each variable, ultimately improving the overall model performance. $A^{\*}$ and $x^{\*}$ denote the corresponding expected solution results.
>
> Reference:
>
> [1] Zhao Q, Wang H, Yue Z, Meng D. A deep variational Bayesian framework for blind image deblurring. Knowledge-Based Systems. 5, 2022, 249:109008.

---

> > ### Comment · Reviewer_vHqD · 2024-11-26
> >
> > Thank you for the detailed feedback and pointing me to Fig 14, which addresses the most serious of my concerns.
> >
> > I'm still not convinced you're actually performing joint MAP estimation and it's non-obvious why (8a) can ignore the likelihood term that depends on A. I also think the notation for A in (8b) needs to change to emphasize an estimate of A is being used in that step.
> >
> > My interpretation of (7-8) is that the method is forming an estimate of A and using that to form an estimate of x. In general that procedure will not produce a joint MAP estimate; this point should be clarified before publication.
> >
> > I have updated my rating.

---

> > > ### Author Response · Authors · 2024-12-03
> > >
> > > We greatly appreciate your thoughtful insights. We would like to emphasize that the splitting strategy for Eq. (7) is outlined in the references [1] we have provided, and our experimental results further validate this design. Therefore, we respectfully ask for your confidence in the correctness of our method.
> > >
> > > We sincerely appreciate your thoughtful and insightful feedback. We would like to kindly highlight that the splitting strategy for Eq. (7) is outlined in the  ref. [1] we provided, and our experimental results offer additional support for the validity of this design. We respectfully hope this clarifies the matter and that you can have confidence in the robustness of our method. Thank you again for your valuable input, and we will address these points to improve the clarity of our work before publication. I appreciate the time and thought you've put into your review and am grateful for the updated rating.
> > >
> > >
> > > Reference:
> > > [1] Zhao Q, Wang H, Yue Z, Meng D. A deep variational Bayesian framework for blind image deblurring. Knowledge-Based Systems. 5, 2022, 249:109008.

---

> ### Author Response · Authors · 2024-11-25
>
> **Q3: Presentation**
>
> We sincerely thank you for your thorough feedback on the presentation. We address each point one by one:
>
> - **About the equation in Line 318**. This equation represents a simplified expression of Eq.(3) after being reformulated in a networked form, allowing for better compatibility with deep learning models.
> - **About that the paper seems to be citing the latest paper to mention a result, rather than the paper where it was first introduced**.  Thanks for your comment. Lensless imaging has emerged as a key focus in computational imaging, with convolutional models serving as a widely adopted framework. Our work extends this paradigm while addressing a critical gap: the model mismatch error ($M^2E$). While previous studies (as discussed in the introduction) have acknowledged $M^2E$, they have not explicitly modeled or mitigated its impact. We are the first to introduce a model for $M^2E$, significantly improving reconstruction accuracy. Experimental results on various lensless imaging prototypes (PHlatCam，FinCam，even DiffuserCam) demonstrate the enhanced generalization and robustness of our method, validating its effectiveness across different setups.
> - **About the meaning of mine**. We sincerely appreciate your question regarding the term "mine". In the context of our work, "mine" refers to the process of "learning" or "extracting" meaningful patterns or information. Specifically, in our M²ER framework, it denotes the process of learning the model mismatch error by leveraging the designed architecture and training strategy. This terminology aligns with common usage in machine learning, where "mining" often refers to uncovering useful insights or features.
>
> **Questions:**
> **Q1:About image Reconstruction in natural scene and proposed method working on the webcam captures and other real-world scenes found in the phlatcam and flatnet papers.** We greatly appreciate your concerns regarding the practical applicability of our method. As shown in  **Fig. 14 (Page 18)** , we have provided the reconstruction results of real-world natural scenes captured by our FinCam. These results demonstrate the ability of our method to generalize effectively to complex 3D environments.
>
> Furthermore, we agree that the same experiments should be conducted with both PHlatCam and FlatNet for a fair comparison, while due to the absence of the corresponding original data provided by the authors, we are unable to verify it. Nevertheless, we acknowledge the importance of such validation and plan to incorporate it in future work to comprehensively assess our method's adaptability to various real-world scenarios, including webcam captures.
>
> **Q2:Would the proposed method work with diffusercams?** We sincerely appreciate your interest in exploring the adaptability of our method. Based on our experiments, our approach is indeed compatible with DiffuserCam systems. Using the dataset from reference [2] and adhering to its data configuration protocols, we conducted a comparative analysis with existing methods, including MMCN [3], FlatNet [4], MWDNs [5], and MDGAN [6]. The results highlight our method's ability to recover detailed scene information effectively, demonstrating its applicability to DiffuserCam setups.
>
> To provide comprehensive evidence, we have included these experiments in the revised manuscript ( **Appendix A.7 and Fig. 17** ). We remain committed to ensuring our method’s applicability across diverse optical imaging systems.
>
> Reference:
>
> [2] K. Monakhova, J. Yurtsever, G. Kuo, N. Antipa, K. Yanny, and L. Waller. Learned reconstructions for practical mask-based lensless imaging. Opt. Express, vol. 27, no. 20, pp. 28075-28090, 2019.
>
> [3] Tianjiao Zeng and Edmund Y. Lam. Robust reconstruction with deep learning to handle model mismatch in lensless imaging. IEEE Transactions on Computational Imaging, vol.7, 1080–1092, 2021
>
> [4] Khan Salman, Siddique, Sundar Varun, Boominathan Vivek,Veeraraghavan Ashok, and Mitra Kaushik. FlatNet: Towards photorealistic scene reconstruction from lensless measurements. IEEE Transactions on Pattern Analysis and Machine Intelligence, vol. 44, no.4, pp.1934-1948, 2022.
>
> [5] Ying Li, Zhengdai Li, Kaiyu Chen, Youming Guo, and Changhui Rao. MWDNs: reconstruction in multi-scale feature spaces for lensless imaging. Opt. Express, vol. 31, no.23, pp. 39088-39101, 2023.
>
> [6] Cong Ni, Chen Yang, Xinye Zhang, Yusen Li, Wenwen Zhang, Yusheng Zhai, Weiji He, and Qian Chen. Address model mismatch and defocus in FZA lensless imaging via model-driven cyclegan. Opt. Lett., vol. 49, no. 15, pp. 4170-4173, 2024.

---

### Note · Authors · 2025-03-06

I have read and agree with the venue's withdrawal policy on behalf of myself and my co-authors.

---

### Meta-Review · Area_Chair_rVtZ · 2024-12-21

**Metareview:**

**Summary.**

This paper propose a joint Maximum a Posteriori (MAP) approach to simultaneously estimate model mismatch error (M2E) and reconstruct high-resolution images from lensless imaging measurements. The proposed method is tested on existing datasets and new dataset captured by the authors using their own lensless camera.

**Strengths.**

Experiments demonstrate excellent performance of the proposed method.

**Weaknesses.**

The proposed recovery method seems similar to existing methods for lensless imaging (e.g., trained denoisers + L1 regularization), but the reconstructed images are almost same as ground truth. This raised flags for the reviewers. The submission did not include the code or dataset to validate this.


The proposed method was trained and tested on a subset of the phlatcam dataset (images presented on a display) where there is relatively little model mismatch. The new dataset was also restricted 2D images presented on a display at a fixed distance from the sensor. There is no evidence the proposed method generalizes to real-world 3D scenes.
Robustness to PSF variations makes sense if the proposed method can handle the variations at test time. It is unclear if the proposed method can handle such variations. It seems to me that the proposed method assumes PSF is fixed for the given dataset and performs all the experiments with that PSF.

The paper contains many (generally minor) mathematical errors.

**Missing.**

The paper is missing experiments that clearly demonstrate that the proposed method is robust to model mismatch.
The paper lacks a clear explanation of how the proposed method differs from existing joint reconstruction and calibration methods.
The submission did not provide code or dataset, which hinders the reproducibility.

**Reasons.**

The main reason for my decision is that the paper did not provide convincing experiments that the proposed method is robust to model mismatch. The reconstructed images are almost same as ground truth, which is physically impossible (primarily due to diffraction blur). The paper did not adequately explain the setting, nor did it include code/dataset to validate the results.

**Additional Comments On Reviewer Discussion:**

The paper had some good discussion among reviewers and authors.

AC and reviewers also discussed the strengths and weaknesses of the paper.

The main concerns of the reviewers were on the mathematical correctness of the derivations and typos, experiments with real scenes (instead of display captures), and lack of experiments that demonstrate robustness to model mismatch. AC and reviewers also discussed the quality of results and how they appear physically infeasible (primarily due to diffraction blur and non-idealities in the systems). All these concerns contributed to the reject decision.

Authors provided responses to clarify some parts and modified the paper with new results, but they could not convince the reviewers.

---

### Decision · Program_Chairs · 2025-01-22

Reject